# A 'torn bag mechanism' of small extracellular vesicle release via limiting membrane rupture of en bloc released amphisomes (amphiectosomes)

Tamás Visnovitz[1,2]*[†], Dorina Lenzinger[1†], Anna Koncz[1,3†], Péter M Vizi[1], Tünde Bárkai[1], Krisztina V Vukman[1], Alicia Galinsoga[1], Krisztina Németh[1,3], Kelsey Fletcher[1], Zsolt I Komlósi[1], Csaba Cserép[4], Ádám Dénes[4], Péter Lőrincz[5], Gábor Valcz[1,3,6], Edit I Buzas[1,3,7]*

[1]Semmelweis University, Department of Genetics, Cell- and Immunobiology, Budapest, Hungary; [2]ELTE Eötvös Loránd University, Department of Plant Physiology and Molecular Plant Biology, Budapest, Hungary; [3]HUN-REN-SU Translational Extracellular Vesicle Research Group, Budapest, Hungary; [4]Laboratory of Neuroimmunology, HUN-REN Institute of Experimental Medicine, Budapest, Hungary; [5]ELTE Eötvös Loránd University, Department of Anatomy, Cell and Developmental Biology, Budapest, Hungary; [6]Department of Image Analysis, 3DHISTECH Ltd, Budapest, Hungary; [7]HCEMM-SU Extracellular Vesicle Research Group, Hungary, Budapest, Hungary

*For correspondence:
visnovitz.tamas@semmelweis.
hu (TV);
buzas.edit@semmelweis.hu (EIB)

[†]These authors contributed equally to this work

## eLife Assessment

In this study, the authors present **compelling** data illustrating a potential mechanism for a hitherto not described form of extracellular vesicle biogenesis. Their model suggests that small extracellular vesicles are secreted from cells within larger vesicles, termed amphiectosomes, which subsequently rupture to release their smaller vesicle contents. This discovery represents an **important** advancement in the field.

**Abstract** Recent studies showed an unexpected complexity of extracellular vesicle (EV) biogenesis pathways. We previously found evidence that human colorectal cancer cells in vivo release large multivesicular body-like structures en bloc. Here, we tested whether this large EV type is unique to colorectal cancer cells. We found that all cell types we studied (including different cell lines and cells in their original tissue environment) released multivesicular large EVs (MV-lEVs). We also demonstrated that upon spontaneous rupture of the limiting membrane of the MV-lEVs, their intraluminal vesicles (ILVs) escaped to the extracellular environment by a 'torn bag mechanism'. We proved that the MV-lEVs were released by ectocytosis of amphisomes (hence, we termed them amphiectosomes). Both ILVs of amphiectosomes and small EVs separated from conditioned media were either exclusively CD63 or LC3B positive. According to our model, upon fusion of multivesicular bodies with autophagosomes, fragments of the autophagosomal inner membrane curl up to form LC3B positive ILVs of amphiectosomes, while CD63 positive small EVs are of multivesicular body origin. Our data suggest a novel common release mechanism for small EVs, distinct from the exocytosis of multivesicular bodies or amphisomes, as well as the small ectosome release pathway.

## Introduction

Extracellular vesicles (EVs) are phospholipid bilayer enclosed structures (*Buzas, 2023*; *György et al., 2011*; *Théry et al., 2018*; *Welsh et al., 2024*), which have important roles in cellular homeostasis and intercellular communication. Exosomes have been defined as small (~50–200 nm) EVs (sEVs) of endosomal origin (*Buzas, 2023*; *Théry et al., 2018*; *Welsh et al., 2024*). Although autophagy is a major cellular homeostatic mechanism and is implicated in a broad spectrum of human diseases, the intersection of autophagy and exosome secretion remains poorly understood. Recently, regulatory interactions have been shown between autophagy-related molecules and EV biogenesis (*Guo et al., 2017*; *Murrow et al., 2015*). Furthermore, the LC3-conjugation machinery was demonstrated to specify the cargo packaged into EVs (*Leidal et al., 2020*). Importantly, both others and we reported the secretion of LC3-carrying exosomes (*Leidal et al., 2020*; *Minakaki et al., 2018*). Particularly relevant to the findings presented here is the implication of amphisomes hybrid organelles formed by the fusion of late endosomes/multivesicular bodies (MVBs) with autophagosomes (*Berg et al., 1998*; *Fader et al., 2008*) in EV biogenesis. It was suggested that fusion of the limiting membrane of amphisomes with the plasma membrane of cells results in a subsequent release of exosomes by exocytosis (*Buzas, 2023*; *Théry et al., 2018*; *Jeppesen et al., 2019*). The current study was prompted by our recent data showing the in vivo en bloc release of large, MVB-like sEV clusters by human colorectal cancer cells (*Valcz et al., 2019*). Here, we investigated if this was a colorectal cancer cell-specific phenomenon. Unexpectedly, we found that it was a general mechanism of sEV release that we designated as 'torn bag mechanism'.

## Results and discussion

In this study, we analyzed in situ fixed, cultured cells with the released EVs preserved in their original microenvironment on a surface coated by gelatin and fibronectin. We detected large multivesicular EVs (MV-lEVs) in sections of different immersion fixed organs. We tested tumorous HT29, HepG2, and non-tumorous HEK293, HEK293T-PalmGFP, HL1 cell lines, as well as primary suspension-type bone marrow-derived mast cells (BMMCs). In addition, we studied ultrathin sections of mouse kidney and liver.

By the analysis of transmission electron micrographs of all tested cell types, we identified budding (*Figure 1A–G*) and secretion (*Figure 1H–N*) of MV-lEVs carrying ILVs. Importantly, in all cases we found evidence for the extracellular rupture of the limiting membrane of MV-lEVs and the release of ILVs (*Figure 1O–U*). For this novel type of sEV release, we suggest the designation 'torn bag mechanism', which is distinct from the exocytosis of MVBs and amphisomes (*Buzas, 2023*; *Théry et al., 2018*; *Welsh et al., 2024*; *Jeppesen et al., 2019*) and from the release of plasma membrane-derived sEVs by ectocytosis (*Mathieu et al., 2021*).

Most relevant to the in vivo conditions, we also observed the same phenomenon within the ultrathin sections of both murine kidney (*Figure 1V*) and liver (*Figure 1W and X*). In these cases, both the intact MV-lEVs (*Figure 1V–X*) and the 'torn bag release' of sEVs (*Figure 1V*) were detected. *Figure 1X* shows that a circulating leukocyte releases MV-lEVs by ectocytosis. In *Figure 1—figure supplement 1*, MV-lEVs, lEVs, and sEVs were captured simultaneously in both kidney (*Figure 1—figure supplement 1C and D*) and liver (*Figure 1—figure supplement 1F, H, I*). In the mouse liver section (*Figure 1—figure supplement 1G*), MV-lEV secretion by both endothelial and subendothelial cells can be detected.

Based on the transmission electron microscopy (TEM) analysis of ultrathin sections, it was not always obvious whether the secreted MV-lEVs had a single or double membrane. However, several micrographs suggested an at least partially intact double membrane (*Figure 1Y-AF*) of MV-lEVs. In the case of BMMCs (*Figure 1Y*), the release phase of a multivesicular structure is captured. The bottom portion of this structure, embedded in the cytoplasm, is surrounded by a single membrane while the upper (budding) portion is covered by double membrane. We hypothesize that disruption of the original amphisome membrane mainly occurs after separation of the MV-lEV from the cell to avoid the release of ILVs inside the cell.

Next, we decided to further investigate the subcellular origin of the ILVs within the secreted MV-lEVs. First, we analyzed the microenvironment of in situ fixed HEK293T-PalmGFP cells by confocal microscopy. The PalmGFP signal of HEK293T-PalmGFP cells principally associates with the plasma

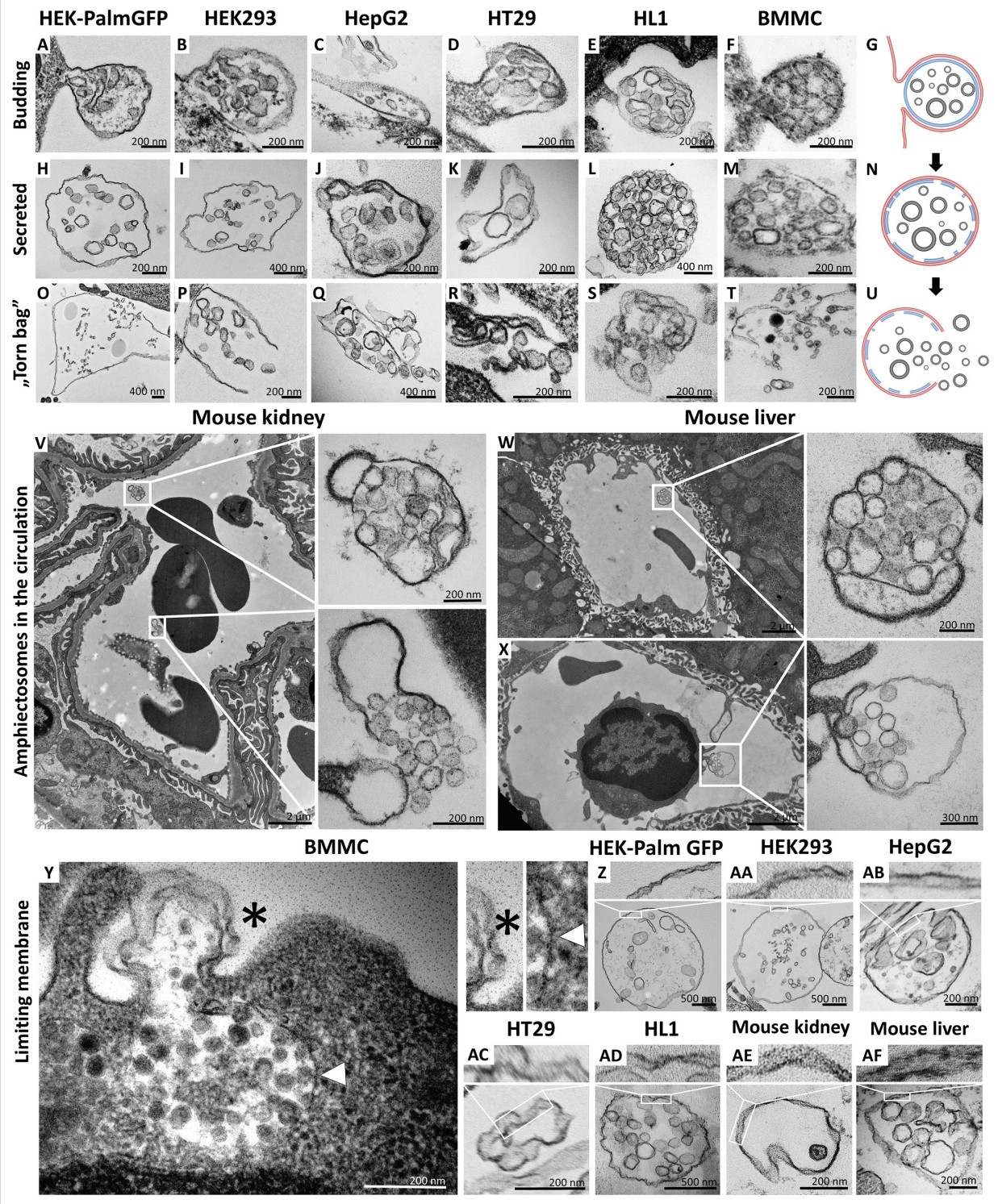

**Figure 1.** Transmission electron microscopic detection of the release and extracellular fate of large, multivesicular extracellular vesicles (MV-lEVs) secreted by different cell lines and cells in mouse organs. Major steps of the release of MV-lEVs were detected in the case of all tested cell lines including the immortal, non-tumorous HEK293T-PalmGFP (**A, H, O**), HEK293 (**B, I, P**), the tumorous cell lines HepG2 (**C, J, Q**) and HT29 (**D, K, R**), the beating cardiomyocyte cell line HL1 (**E, L, S**) and the primary suspension of bone marrow-derived mast cells (BMMCs) (**F, M, T**). The different phases of EV secretion were also captured in the circulation of mouse kidney (**V**) and liver (**W, X**). According to the electron micrographs, we found evidence for the budding (**A–G, X**) and secretion (**H–N, V, W**) of the MV-lEVs. We also detected the extracellular rupture of the limiting membrane of the released MV-lEVs with the escape of the intraluminal vesicles (ILVs) by a 'torn bag mechanism' (**O–U, V**). Although it is not always clear whether the secreted MV-

*Figure 1 continued on next page*

*Figure 1 continued*

lEVs have a single or double limiting membrane, several micrographs suggest the presence of the double membrane (**Y–AF**) in the secreted MV-lEVs. In the case of BMMCs (**Y**), the release phase of a multivesicular structure is captured. The bottom portion of this structure embedded in the cytoplasm is surrounded by a single membrane (white arrowhead) while the upper (budding) portion is covered by double membrane (asterisk). In the schematic figures (**G, N, U**) the limiting membrane of MV-lEV presumably with plasma membrane origin was indicated by red, the original limiting membrane of intracellular amphisomes, which may be fragmented during the release process was indicated by blue while the ILVs of the MV-lEV were shown by gray color. Panel G was created with BioRender.com. Panel N was created with BioRender.com. Panel U was created with BioRender.com.

The online version of this article includes the following figure supplement(s) for figure 1:

**Figure supplement 1.** Additional transmission electron micrographs of mouse kidney and liver sections.

membrane (**Kovács et al., 2023**; **Figure 2—figure supplement 1A–C**), therefore the green fluorescence helped us to identify the plasma membrane-derived limiting membrane of MV-lEVs. In agreement with our previous findings on HT29 colorectal cancer cells, within the MV-lEVs, we found CD63/ ALIX (**Figure 2A and G**), CD81/ALIX (**Figure 2B and H**), CD63/TSG101 (**Figure 2C and I**), and CD81/ TSG101 (**Figure 2D and J**) double positive ILVs or ILV clusters.

We also studied the possible autophagy-related aspects of the secreted MV-lEVs. ILVs were tested for the autophagy marker LC3B in parallel with CD63 and CD81. Although LC3B, CD63, and CD81 were all present in association with the ILVs (**Figure 2E and F**), the LC3B and CD63 (**Figure 2K**) and the LC3B and CD81 (**Figure 2L**) signals did not overlap. **Figure 2M** shows that while the known sEV markers (CD63, CD81, TSG101, and ALIX) strongly co-localized with each other, LC3B positivity hardly showed co-localization with CD63 or CD81. Immunocytochemistry analysis of HT29, HepG2, and the cardiomyoblast H9c2 cells further validated the findings obtained with the HEK293T-PalmGFP cells (**Figure 2—figure supplements 2 and 3**). The ILVs of HEK293T-PalmGFP and HepG2 cell lines were also Rab7 positive (**Figure 2—figure supplement 3A and B**), suggesting a late endosomal origin. Western blotting of the applied antibodies is summarized in **Figure 2—figure supplement 4**.

The sEV markers were also tested by TEM using negative-positive contrasting technique (**Théry et al., 2006**) on sEVs separated form serum-free conditioned medium of HEK293T-PalmGFP cells. **Figure 2P** confirms the typical sEV morphology. With TEM double immunogold labeling, using anti-LC3B and anti-CD63 antibodies simultaneously, we found distinct LC3B positive (**Figure 2Q**, **Figure 2—figure supplement 6O**) and CD63 positive (**Figure 2R**, **Figure 2—figure supplement 6O**) sEVs. Based on the analysis of TEM images, the diameters of unlabeled and LC3B positive and negative sEVs were determined (**Figure 2S**). The LC3B positive sEVs had a significantly larger diameter as compared to the LC3B negative ones.

To conclude our marker studies, we detected the presence of CD63, CD81, TSG101, ALIX positive, most probably MVB-derived ILVs. In addition, the autophagosome marker carrying LC3B positive ILVs were also found within the same single, plasma membrane limited extracellular MV-lEV, which identified these MV-lEVs as en bloc released amphisomes (**Klionsky, 2021**) that we refer to 'amphiectosomes'.

The 'torn bag mechanism' was also monitored by live-cell SIM$^2$ super-resolution microscopy analysis of HEK293T-PalmGFP-LC3RFP cells (**Figure 2N and O**). The release of the LC3 positive red fluorescent signal was detected within a relatively short period of time (the first LC3 positive ILVs left the amphiectosome within 40 s, the whole 'torn bag' sEV release process was completed within 260 s) (**Figure 2O**). We could rule out the possibility that rupture of the limiting membrane detected by TEM (**Figure 1O–T, V**) was a fixation artifact by showing the spontaneous release of LC3 positive sEVs from amphiectosomes with live-cell imaging. Characterization of the in-house developed HEK293T-PalmGFP-LC3RFP cell line is shown in **Figure 2—figure supplement 5**.

In the following step, we addressed the question whether LC3, associated with the ILVs of MV-lEVs, indeed reflected autophagy origin. We tested MVBs (**Figure 2—figure supplement 6A, F, and K**), autophagosomes (**Figure 2—figure supplement 6B, G, and L**), amphisomes (**Figure 2— figure supplement 6C, H, and M**), amphiectosomes (**Figure 2—figure supplement 6D, I, and N**), and isolated sEV fractions of the same cells (**Figure 2—figure supplement 6E, J, and O**). Using immune electron microscopy, as expected, we found CD63 single positivity in MVBs (**Figure 2— figure supplement 6K**). In autophagosomes, the limiting phagophore membrane was LC3B positive, and CD63 positivity was also present (**Figure 2—figure supplement 6L**). The limiting membrane of amphisomes was LC3B negative, and the internal membranous structures were either LC3B or CD63

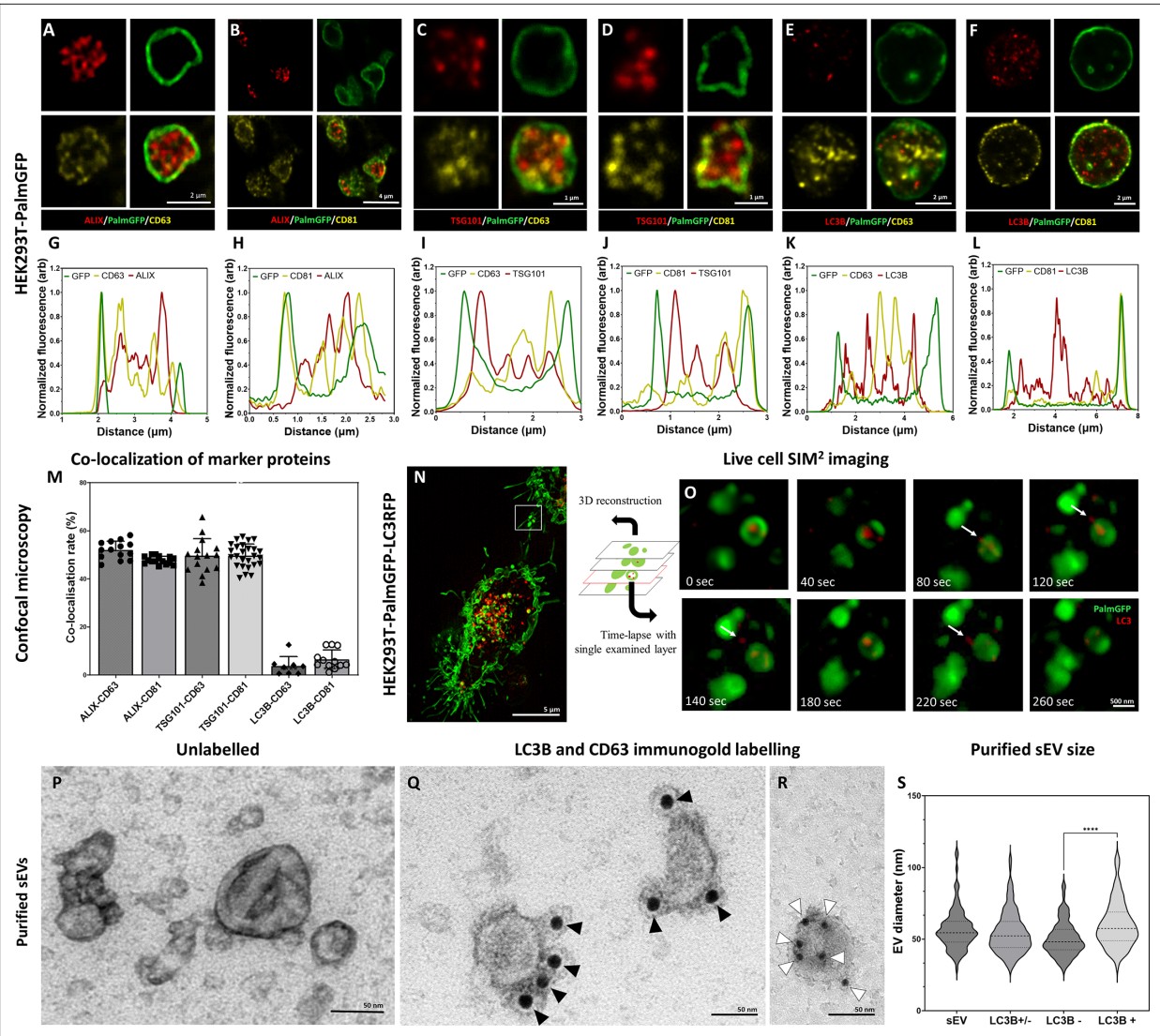

**Figure 2.** Detection of conventional small extracellular vesicle (sEV) markers and the LC3 protein in HEK293T-PalmGFP cell-derived EVs. Widely used sEV markers (CD63, CD81, ALIX, and TSG101) and LC3B were tested in multivesicular large EVs (MV-lEVs) found in the microenvironment of the releasing cells by confocal microscopy after in situ fixation (**A–F**). Normalized fluorescence intensities were calculated to determine the relative localization of the limiting membrane (PalmGFP), the conventional sEV markers, and the LC3B signal (**G–L**). Fluorescence intensity peaks of sEV markers were largely overlapping with each other, while the LC3B signal and the sEV markers showed separation. Co-localization rates were also calculated (**M**). The sEV markers co-localized with one another as no significant difference was found among them. In contrast, low co-localization rates were detected between the 'classical' sEV markers and LC3B (one-way ANOVA, p<0.0001, n=8-26 confocal images). Error bars represent standard deviation. Real-time release of LC3 positive sEVs by the 'torn bag mechanism' was studied in the case of HEK293T-PalmGFP-LC3RFP cells by Elyra7 SIM² super-resolution live-cell imaging (**N,O**). Images were recorded continuously and selected serial time points are shown. LC3 positive, red fluorescent small particles were released within a 5 min timeframe (**O**) and are indicated by white arrows. Presence of CD63 and LC3B were detected in the case of an sEV fraction separated from serum-free condition medium using immunogold transmission electron microscopy (TEM). HEK293T-PalmGFP-derived sEV fraction is shown by negative-positive contrast without immune labeling (**P**). In double-labeled immunogold TEM images (**Q, R**), distinct LC3B positive (**Q**) and CD63 positive (**R**) sEVs were found. However, CD63-LC3B double positive EVs were not detected. Black arrowheads indicate 10 nm gold particles identifying LC3B, while white arrowheads show 5 nm gold particles corresponding to the presence of CD63. Quantitative analysis of TEM images was performed (**S**), and the diameters of different EV populations were determined. The LC3B negative population was significantly smaller than the LC3B positive one (p<0.0001, t-test; n=79–100). No difference was detected when the immunogold labeled sEV fraction (either LC3B positive or negative, LC3B+/-) and the unlabeled sEV fraction (sEV) were compared (p<0.05, t-test, n=112–179).

The online version of this article includes the following source data and figure supplement(s) for figure 2:

**Source data 1.** XLSX file containing data points of **Figure 2G, H, I, J, K, L, M, and S**.

**Figure supplement 1.** Localization of GFP signal in HEK293T-PalmGFP cells.

*Figure 2 continued on next page*

*Figure 2 continued*

**Figure supplement 1—source data 1.** XLSX file containing data points of *Figure 2-figure supplement 1C*.

**Figure supplement 2.** Confocal microscopic images of amphiectosome release by HT29 and HepG2 cells.

**Figure supplement 2—source data 1.** XLSX file containing data points of *Figure 2—figure supplement 2I*.

**Figure supplement 3.** Additional confocal microscopic images of H9c2, HEK293T-PalmGFP, and HepG2 cells-derived multivesicular large extracellular vesicles (MV-lEVs).

**Figure supplement 4.** Qualitative western blot validation of antibodies used in immunofluorescence detection.

**Figure supplement 4—source data 1.** Original files for western blot analysis displayed in *Figure 2—figure supplement 4*.

**Figure supplement 4—source data 2.** Original western blots for *Figure 2—figure supplement 4*, indicating the relevant bands and cell lines.

**Figure supplement 5.** Characterization of the in-house generated HEK293T-PalmGFP-LC3RFP cell line.

**Figure supplement 5—source data 1.** Original files for western blot analysis displayed in *Figure 2—figure supplement 5B*.

**Figure supplement 5—source data 2.** Original western blots for *Figure 2—figure supplement 5B*, indicating the relevant bands, cell lines, and treatments.

**Figure supplement 6.** Structures involved in amphiectosome release.

**Figure supplement 6—source data 1.** XLSX file containing data points of *Figure 2—figure supplement 6Q and R*.

positive (*Figure 2—figure supplement 6M*). The same immunoreactivity was also observed in the ILVs of the released amphiectosomes (*Figure 2—figure supplement 6N*). Importantly, sEVs separated from serum-free conditioned medium of HEK293T-PalmGFP cells were either LC3B or CD63 positive (*Figure 2—figure supplement 6O*). Thus, we confirmed our confocal microscopy results at the ultrastructural level. Using immunogold TEM, we provided further evidence for the budding/ectocytosis mechanism of amphiectosome release (*Figure 2—figure supplement 6P*). The diameters of ILVs within MVBs, amphisomes, and amphiectosomes were compared (*Figure 2—figure supplement 6Q*), and the differences were likely due to the different membrane composition, pH, and osmotic conditions within these structures. In agreement with our observations with separated sEVs, LC3B positive ILVs had a significantly larger diameter than the LC3B negative ones (*Figure 2—figure supplement 6R*) possibly indicating difference in membrane composition and their different intracellular origin. Based on all the above findings, we propose the following model (*Figure 3A*): autophagosomes and MVBs fuse to form amphisomes, and the inner, LC3 positive membrane of autophagosomes undergoes fragmentation (*Klionsky, 2021*). Membrane fragments curl up and form LC3 positive ILVs. Therefore, amphisomes contain both MVB-derived CD63 positive/LC3 negative and autophagosome-derived, CD63 negative/LC3 positive ILVs. The amphisome is next released from the cell by ectocytosis. Finally, the plasma membrane-derived outer membrane ruptures enabling the ILVs escape to the extracellular space by a 'torn bag mechanism'. By using stimulated emission depletion (STED) microscopy, we documented the intracellular phases of our proposed model: MVB (*Figure 3B*), autophagosome (*Figure 3D*), the fusion of MVB and autophagosome (*Figure 3C*), fragmentation of the LC3 positive membrane and ILV formation from the membrane fragments (*Figure 3F*) and mature amphisome (*Figure 3E*). The plasma membrane origin of the external membrane of amphiectosome was further supported by wheat germ agglutinin (WGA)-based live-cell labeling (*Figure 3G*).

To investigate the process of amphiectosome release, we exposed the MV-lEV releasing cells to different in vitro treatments (*Figure 3H*). The release of MV-lEVs was monitored by confocal microscopy of in situ fixed cell cultures. Optimal test conditions were determined (*Figure 3—figure supplement 1A–F*) and the results are summarized in *Figure 3I*. Original LASX files which served as a basis of our quantification are publicly available (doi: 10.6019/S-BIAD1456). An example for our approach to count the MV-lEVs is shown in *Figure 3—figure supplement 1H*. Cytochalasin B did not have any effect on the discharge of MV-lEVs suggesting that the release did not involve a major actin-dependent mechanism. In contrast, there was a significant reduction of the MV-lEV secretion upon exposure of the cells to Colchicine indicating a role of microtubules in the release of the MV-lEVs. While Rapamycin significantly reduced the discharge of MV-lEVs, Chloroquine and Bafilomycin induced an enhanced MV-lEV secretion. Rapamycin activates autophagic degradation (*Xie et al., 2021*), therefore, it induces a shift toward degradation as opposed to secretion. The lysosomotropic agents Chloroquine and Bafilomycin are known to interfere with the acidification of lysosomes (*Chen et al., 2011*; *Wang et al., 2021*). By blocking the degradation pathway of MVBs/amphisomes (*Figure 3H*), an

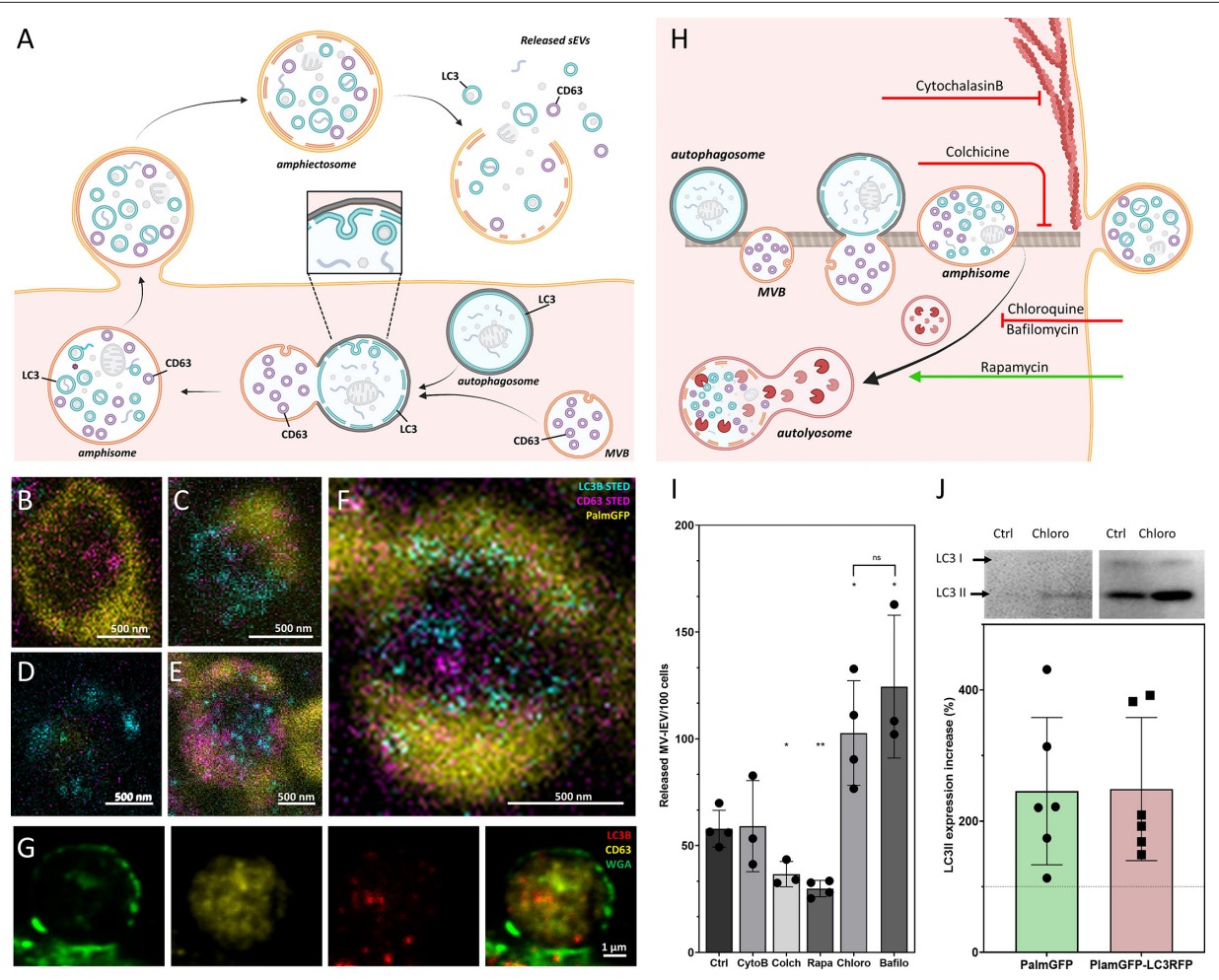

**Figure 3.** Amphiectosome release and its modulation. Based on our data, a model of amphiectosome release was generated (**A**). Panel A was created with BioRender.com. According to this model, the fusion of multivesicular bodies (MVBs) and autophagosomes forms amphisomes. The LC3B positive membrane layer (indicated in cyan) undergoes disintegration and forms LC3B positive intraluminal vesicles (ILVs) inside the amphisome. Later, the amphisome is released into the extracellular space by ectocytosis and can be identified extracellularly as an amphiectosome. Finally, the limiting membrane(s) of the amphiectosome is ruptured and the ILVs are released as small extracellular vesicles (sEVs) into the extracellular space by a 'torn bag mechanism'. Steps of amphisome formation including LC3 positive ILV formation in 30 µM Chloroquine-treated HEK293T-PalmGFP cells was followed by super-resolution (stimulated emission depletion [STED]) microscopy (**B–F**). The super-resolution STED channels were LC3B (cyan) and CD63 (magenta), while yellow indicates the confocal PalmGFP signal. Intracellular vesicular structures (such as endosomes, MVBs, and amphisomes) may receive PalmGFP from the plasma membrane. An MVB (**B**), an autophagosome with PalmGFP negative membrane (**D**), fusion of an autophagosome and an MVB (**C**), formation of LC3B positive ILVs in an amphisome (**F**), and a mature amphisome (**E**) were detected. To confirm the origin of the external membrane layer of amphiectosomes, fluorescently labeled wheat germ agglutinin (WGA) was applied. The plasma membrane of the living non-fluorescent HEK293 cells was labeled. As the external membrane of the budding amphiectosome was WGA positive, its plasma membrane origin is confirmed (**G**). To further support our model on amphiectosome release and 'torn bag' EV secretion, different in vitro treatments were applied. Cytochalasin B, Colchicine, Chloroquine, Bafilomycin A1, and Rapamycin were used to modulate amphiectosome release. Targeted molecular processes are summarized (**H**). Panel H was created with BioRender.com. While Cytochalasin B inhibits actin-dependent membrane budding and cell migration, Colchicine blocks the microtubule-dependent intracellular trafficking. While Chloroquine and Bafilomycin have similar, Rapamycin has opposite effect on lysosome-autophagosome or lysosome-amphisome fusion. Chloroquine and Bafilomycin inhibit lysosomal degradation while Rapamycin accelerates it. Based on confocal microscopy, Cytochalasin B (CytoB) did not alter the dynamics of amphiectosome release (**I**). In contrast, both Colchicine (Colch) and Rapamycin (Rapa) significantly inhibited the release of amphiectosomes, while Chloroquine (Chloro) and Bafilomycin (Bafilo) increased the release frequency. There was no difference between the effect of Chloroquine and Bafilomycin (**I**). Results are shown as mean ± SD of three to four independent biological replicates, analyzed by one-way ANOVA and Student's t test, *: p<0.05, **: p<0.01, ns: non-significant. Original LASX files, which served as a basis of our quantification, are publicly available (doi: 10.6019/S-BIAD1456). Example for the calculation is shown in *Figure 3—figure supplement 1H*. Presence of membrane-bound (lipidated) LC3II was tested by western blotting. The total protein content of serum-, cell-, and large EV-depleted conditioned medium of HEK293T-PalmGFP (PalmGFP) and HEK293T-PalmGFP-LC3RFP (PalmGFP-LC3RFP) cells was precipitated by TCA and 20 µg of the protein samples were loaded on the gel (**J**). The lipidated LC3II band was detected in all cases. Relative expression of control (Ctrl) and Chloroquine

*Figure 3 continued on next page*

*Figure 3 continued*

(Chloro)-treated samples were determined by densitometry. Chloroquine treatment increased the LC3II level by approximately twofold. Results are shown as mean ± SD of n=6 biological replicates.

The online version of this article includes the following source data and figure supplement(s) for figure 3:

**Source data 1.** XLSX file containing data points of *Figure 2I and J*.

**Source data 2.** Original files for western blot analysis displayed in *Figure 3J*.

**Source data 3.** Original western blots for *Figure 3J*, indicating the relevant bands, cell lines, and treatments.

**Figure supplement 1.** Supporting information for treatments and size distribution of multivesicular large extracellular vesicles (MV-lEVs).

---

enhanced sEV secretion is observed. This effect is well known for exosome secretion from MVBs (*Edgar et al., 2016*; *Ortega et al., 2019*). The diameters of the released MV-lEVs were determined based on confocal images (*Figure 3—figure supplement 1G*). Metabolic activity of the cells was determined by a Resazurin assay, and a significant reduction was detected upon exposure of the cells to Rapamycin (*Figure 3—figure supplement 1D*) in line with previously published data (*Zhang et al., 2020*). LC3II is the membrane-associated, lipidated autophagic form of LC3 (*Tanida et al., 2008*) and it is the hallmark of autophagy-related membranes (*Klionsky, 2021*). Importantly, by western blot, we not only showed the presence of the membrane-bound LC3II in serum-free, lEV-depleted (sEV containing) conditioned medium of both HEK293T-PalmGFP and HEK293T-PalmGFP-LC3RFP cells, but the amount of LC3II substantially increased upon Chloroquine treatment (*Figure 3J*). Raw data of western blots are available in *Figure 3—source data 2* and *Figure 3—source data 3*.

Recent advances in the EV field shed light on migrasomes, a special type of MV-lEVs (*Liang et al., 2023*; *Ma et al., 2015*). With their pomegranate-like ultrastructure, migrasomes resemble amphiectosomes. Therefore, we tested the presence of TSPAN4, a migrasome limiting membrane marker (*Ma et al., 2015*), in amphiectosomes. *Figure 4A, B, G, and H* shows that although TSPAN4 was present intraluminally in the HEK293T-PalmGFP-derived MV-lEVs, it was clearly absent from their external membrane. Surprisingly, we identified two different HT29 cell-derived MV-lEV populations: one in which TSPAN4 was only located intraluminally (*Figure 4C, E, I, and K*), and another one with a TSPAN4 positive external membrane (*Figure 4D, F, J, and L*). This raised the possibility that the latter population corresponded to migrasomes. Our co-localization analysis also confirmed the existence of two distinct MV-lEV populations (*Figure 4M*). Next, we carried out live-cell imaging on HEK293T-PalmGFP-LC3RFP cells. The released MV-lEVs were either LC3 positive or negative intraluminally (*Figure 4N and O*). Our TEM images confirmed that certain cell types can release both migrasome-like structures and amphiectosomes. MV-lEVs with typical migrasome-associated retraction fiber(s) were detected in the case of HL1 (*Figure 4P*), HEK293T-PalmGFP (*Figure 4Q*), and BMMC cells (*Figure 4R*). Of note, it cannot be excluded that the elongated structures observed in the above cases may correspond to tunnelling nanotubes (*Drab et al., 2019*). Importantly, the same cell lines also released amphiectosomes by budding from the cell surface (*Figure 4S–U*). Taken together, based on the absence of TSPAN4 in their external membrane, and their lack of association with retraction fibers, amphiectosomes appear to be distinct from migrasomes. Besides migrasomes, another MV-lEV type was described in the case of gastrointestinal tumors and low-grade glioblastoma cells referred to as spheresome (*Baselga et al., 2023*; *Junquera et al., 2016*). However, there is no data on a relationship of spheresome release and autophagy. Recently, endothelial cell-derived, multicompartmented microvesicles (MCMVs) were shown to protrude and pinch off from the cell surface releasing ILVs by a mechanism similar to exocytosis (*Petersen et al., 2023*). The absence of protrusion clusters described for MCMV (*Petersen et al., 2023*) distinguishes amphiectosomes from MCMVs. In addition, the previously described so-called 'nodal vesicular parcels' (*Tanaka et al., 2005*) might be special examples of amphiectosomes. Finally, in *Caenorhabditis elegans*, the release autophagy and stress-related large EVs (lEVs) (exophers) has been documented (*Melentijevic et al., 2017*; *Cooper et al., 2021*; *Yang et al., 2024*). They contain damaged organelles and do not have an MV-lEV-like ultrastructure. In contrast, the amphiectosomes we described here have multivesicular structure without recognizable damaged organelles.

Our approach, involving in situ fixation of cultures and tissues, made it possible to recognize sEV release from amphiectosomes by the 'torn bag mechanism'. We propose that this mechanism could be easily missed earlier if conditioned medium was subjected to centrifugation, SEC purification, or

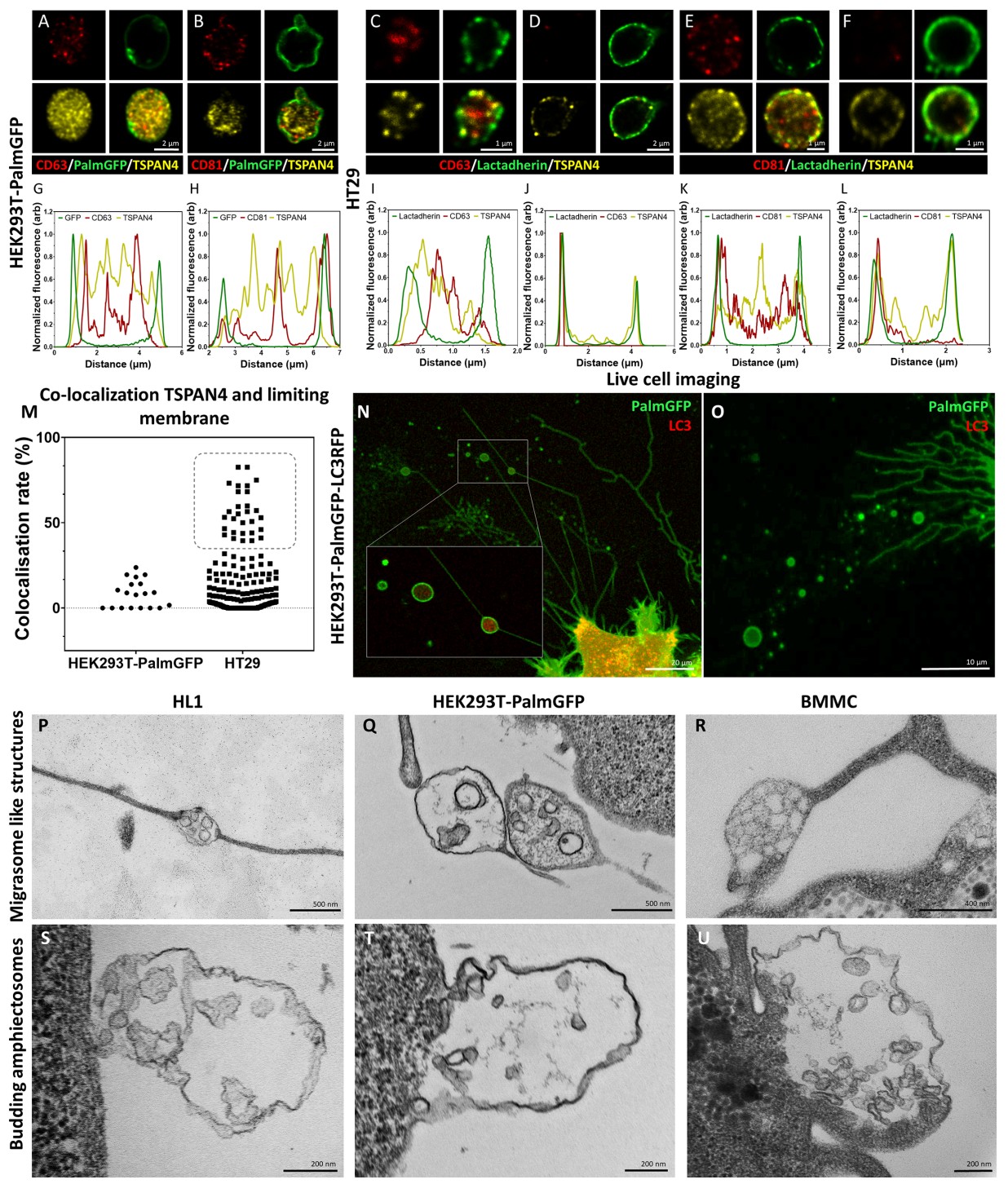

**Figure 4.** Comparison of amphiectosomes and migrasomes. Commonly used small extracellular vesicle (sEV) markers (CD63, CD81) and TSPAN4, a suggested migrasome marker, were tested in in situ fixed intact multivesicular large EVs (MV-lEVs) of HEK293T-PalmGFP (**A, B**) and HT29 (**C–F**) cells by confocal microscopy. Normalized fluorescence intensities were calculated to determine the relative localization of the limiting membrane (with PalmGFP or lactadherin labeling) and the CD63/TSPAN4 and CD81/TSPAN4 markers (**G–L**). In the case of HEK293T-PalmGFP-derived EVs, we did not find migrasomes with TSPAN4 in their limiting membrane. The TSPAN4 signal was only detected intraluminally in the MV-lEVs. The limiting membranes of HT29-derived MV-lEVs were either TSPAN4 positive or negative. The co-localization rate between the limiting membrane and TSPAN4 was low in case of HEK293T-PalmGFP-derived EVs. In the case of HT29 cells, two MV-lEV populations were identified: one with low and one with high co-localization rates (**M**). Live-cell imaging of HEK293T-PalmGFP-LC3RFP cells showed retraction fiber-associated MV-lEVs with or without intraluminal LC3 positivity (**N, O**). Using transmission electron microscopy (TEM), we could identify structures with retraction fiber-associated morphology in the case of HL1 cells

*Figure 4 continued on next page*

*Figure 4 continued*

(**P**), HEK293T-PalmGFP cells (**Q**), and bone marrow-derived mast cells (BMMCs) (**R**). For comparison, budding of amphiectosomes of the same HL1 cells (**S**), HEK293TPalmGFP cells (**T**), and BMMCs (**U**) are shown (without being associated with long retractions fibers).

even to simple pipetting, which may rupture the limiting membrane of amphiectosomes. This aligns with our observation that the spontaneous escape of ILVs from untouched amphiectosomes can be completed as early as 5 min after amphiectosome release. Based on our data presented here, and considering that the exocytosis of MVBs/amphisomes under steady-state conditions is rarely documented in the scientific literature, we suggest that amphiectosome secretion and the 'torn bag mechanism' may have a significant, yet previously unrecognized, role in sEV biogenesis.

# Materials and methods

**Key resources table**

| Reagent type (species) or resource | Designation | Source or reference | Identifiers | Additional information |
|---|---|---|---|---|
| Cell line (*Homo sapiens*) | HEK293 human kidney(embryonic) | ECACC (Sigma) | #85120602 RRID:CVCL_0045 | Batch No: 18E026 |
| Cell line (*Homo sapiens*) | HT29 Caucasian colon adenocarcinoma grade II | ECACC (Sigma) | #91072201 RRID:CVCL_0320 | Batch No: 09K003 |
| Cell line (*Homo sapiens*) | HepG2 human hepatocyte carcinoma | ECACC (Sigma) | #85011430 RRID:CVCL_0027 | Batch No: 19B009 |
| Cell line (*Mus musculus*) | HL1 mouse cardiomyocyte cell line, atrial | Merck | # SCC065 RRID:CVCL_0303 | Batch No: RD1601001 |
| Cell line (*Homo sapiens*) | HEK293TPalmGFP human kidney(embryonic) expressing palmitoylated GFP | Kind gift of Charles Lai https://doi.org:10.1038/ncomms8029 | | Resorted before MCB preparation |
| Cell line (*Homo sapiens*) | HEK293T-PalmGFPLC3RFP human kidney(embryonic) expressing palmitoylated GFP and RFP tagged LC3 | This paper | | See Materials and methods |
| Cell line (*Mus musculus*) | BMMC bone marrow-derived mast cells | Primary cell culture https://doi.org/10.1002/jev2.12023 | | |
| Cell line (Rattus norvegicus) | H9C2 (2-1) rat cardiovascular , Myoblast | ECACC (Sigma) | #88092904, RRID:CVCL_0286 | Batch No: 17A028 |
| Transfected construct | LentiBrite RFP-LC3 Lentiviral Biosensor | Merck | 17-10143 | Batch No: 3530171 |
| Biological sample (*Mus musculus*) | own animal house | | C57BL/6 RRID:MGI:2159769 | male, 12 weeks of age |
| Antibody | rabbit polyclonal anti-CD63 (Cterminal) | Sigma/Merck | SAB2109138 | IF (1:200) WB (1:500) |
| Antibody | mouse monoclonal anti-CD63 | Santa Cruz Biotechnology | MX-49.129.5 clone: sc-5275 RRID:AB_627877 | IF (1:200) TEM (1:50) WB (1:1000) |
| Antibody | rabbit polyclonal anti-CD81 | Sigma/Merck | SAB3500454 RRID:AB_10640751 | IF (1:200) WB (1:2500) |
| Antibody | mouse monoclonal anti-CD81 | Invitrogen | MA5-13548 clone: 1.3.3.22 RRID:AB_10987151 | IF (1:100) WB (1:100) |

*Continued on next page*

*Continued*

| Reagent type (species) or resource | Designation | Source or reference | Identifiers | Additional information |
|---|---|---|---|---|
| Antibody | rabbit polyclonal anti-TSG101 | Sigma/Merck | HPA006161 RRID:AB_1080408 | IF (1:200) WB (1:1000) |
| Antibody | rabbit polyclonal anti-ALIX (Cterminal) | Sigma/Merck | SAB420047 | IF (1:200) WB (1:1000) |
| Antibody | rabbit monoclonal anti-LC3B | Sigma/Merck | ZRB100 clone: 12K5 | IF (1:200) TEM (1:50) WB (1:1000) |
| Antibody | rabbit monoclonal anti-LC3A | Sigma/Merck | ZRB1125 clone: 3J12 | IF (1:200) WB (1:1000) |
| Antibody | rabbit polyclonal anti-TSPAN4 | Bioss | BS-9413R | IF (1:200) |
| Antibody | mouse monoclonal anti-Rab7 | Sigma/Merck | R8779 clone: Rab7117 RRID:AB_609910 | IF (1:200) WB (1:1000) |
| Antibody | mouse monoclonal anti-α-tubulin | Sigma/Merck | T9026 clone: DM1A RRID:AB_477593 | IF (1:200) |
| Antibody | mouse monoclonal anti-GFP | Sigma/Merck | G6539 clone: GFP-20 RRID:AB_259941 | IF (1:200) WB (1:1000) |
| Antibody | mouse monoclonal anti-RFP | Invitrogen | MA5-15257 clone: RF5R RRID:AB_10999796 | WB (1:1000) |
| Antibody | goat antimouse IgGATTO550 | Sigma/Merck | 43394 RRID:AB_1137651 | IF (1:1000) |
| Antibody | goat antirabbit IgG-ATTO647N | Sigma/Merck | 40839 RRID:AB_1137669 | IF (1:1000) |
| Antibody | goat antimouse Star 635P | Abberior | ST635P-1001–500UG RRID:AB_2893232 | IF (1:500) |
| Antibody | goat antirabbit Star 580 | Abberior | ST580-1002-500UG RRID:AB_2910107 | IF (1:500) |
| Antibody | goat polyclonal anti-rabbit IgG Fc (HRP) | abcam | ab97200 RRID:AB_10679899 | WB (1:10,000) |
| Antibody | goat polyclonal anti-mouse IgG Fc (HRP) | abcam | ab97265 RRID:AB_10680426 | WB (1:10,000) |
| Antibody | goat polyclonal anti-rabbit IgG (whole molecule) 10 nm gold preadsorbed | abcam | ab27234 RRID:AB_954427 | TEM (1:50) |
| Antibody | goat polyclonal anti-mouse IgG (whole molecule) 5 nm gold preadsorbed | Sigma/Merck | G7527 RRID:AB_259955 | TEM (1:50) |
| Other | CF488A conjugated Wheat Germ Agglutinin (WGA) | Biotium | 29022-1 | Lot Number: 21C0224-1149057 |
| Chemical compound, drug | Bafilomycin A1 | Sigma/Merck | B1793 | Lot Number: 0000190389 |
| Chemical compound, drug | Colchicine | Serva | 77120.02 | Lot Number: 190300 |
| Chemical compound, drug | Chloroquine diphosphate | Invitrogen | P36236 C | Lot Number: 2441325 |
| Chemical compound, drug | Rapamycin | Sigma/Merck | R0395 | Lot Number: 0000084976 |
| Chemical compound, drug | Cytochalasin B | Sigma | C2743 | Lot Number: 037M4083V |
| Chemical compound, drug | FBS | Biosera | FB-1090/500 | Lot Number: 015BS575 |

*Continued on next page*

*Continued*

| Reagent type (species) or resource | Designation | Source or reference | Identifiers | Additional information |
|---|---|---|---|---|
| Other | TFF Easy | HansaBioMed Life Sciences | HBM-TFF/1 | |
| Software, algorithm | LASX | Leica | Leica Application Suite X 3.5.5.19976 | |
| Software, algorithm | ZEN Blue | Zeiss | ZEN 2.3 lite | |
| Software, algorithm | iTEM | Olympus | iTEM 5.1 | |
| Software, algorithm | ImageJ | https://imagej.n et/ij/ | v1.54g | |
| Software, algorithm | Prism9 | GraphPad | GraphPad Prism 9.4.1 | |
| Software, algorithm | BioRender | https://www.biorender.com/ | | |

## Cell lines

The HEK293 human embryonic kidney, the HepG2 human hepatocyte carcinoma cell line, the HT29 human colon adenocarcinoma cell lines, and the H9c2 rat cardiomyoblast cell line were purchased from the European Collection of Authenticated Cell Cultures (ECACC) through their distributor (Sigma). The HL1 cell line was purchased from Millipore. The HEK293TPalmGFP human embryonic kidney cells were kindly provided by *Lai et al., 2015*. Mouse BMMCs were differentiated and expanded as we described previously (*Vukman et al., 2020*). The HEK293, HEK293TPalmGFP, HepG2, and H9c2 cell lines were grown in DMEM (Gibco) (*Koncz et al., 2023*; *Németh et al., 2023*; *Németh et al., 2021*), the HT29 cells were cultured in RPMI 1640 (Gibco) (*Valcz et al., 2019*), while the HL1 cells were grown in Claycomb medium (*Koncz et al., 2023*). All cells were cultured with 10% fetal bovine serum (FBS, BioSera) in the presence of 100 U/mL of penicillin and 100 µg/mL streptomycin (Sigma). Before analysis by confocal microscopy, the cells were cultured on the surface of gelatin-fibronectin-coated glass coverslips (VWR). The coating solution contained 0.02% gelatin (Sigma) and 5 mg/mL fibronectin (Invitrogen). Coverslips were coated overnight (O/N) at 37°C.

To minimize the genetic drift of the cell lines and to ensure consistent quality of the cells, we followed the recommendations of the European Collection of Authenticated Cell Cultures (ECACC). Upon the arrival of the cell lines from an authenticated cell bank, a master cell bank (MCB) was established and subsequently, working cell banks (WCBs) were manufactured and tested. All experiments were initiated using a vial from the WCB.

For TEM, the adherent cells (HEK293, HEK293T-PalmGFP, HepG2, HT29, and HL1) were grown in gelatin-fibronectin-coated eight-well Flux Cell Culture Slides (SPL).

Cell cultures were tested regularly for mycoplasma infection by PCR, with the following PCR primers:

GAAGAWATGCCWTATTTAGAAGATGG and CCRTTTTGACTYTTWCCAC-CMAGTGGTTGTTG (*Koncz et al., 2023*).

## Generation of HEK293T-PalmGFP-LC3RFP cell line

For the generation of a stable HEK293T-PalmGFP-LC3RFP cell line, HEK293T-PalmGFP cells were transfected by LentiBrite RFP-LC3 Lentiviral particles (Merck) according to the instructions of the manufacturer. The GFP-RFP double positive cells were sorted by an HS800 Cell Sorter (SONY), and cell banks (MCB and WCB) were prepared. The success of the stable transfection was analyzed by immunocytochemistry and western blotting. Results are shown in *Figure 2—figure supplement 5*.

## Confocal microscopy

Confocal microscopy was carried out as we described earlier (*Koncz et al., 2023*) with some modifications. As serum starvation significantly affects autophagy (*Wang et al., 2023*), and EV-depleted FBS in the cell culture medium may influence cellular physiology and morphology (*Lehrich et al., 2021*), FBS was not removed before fixation. Our study focuses on lEVs with diameter >350–500 nm. EVs in this size range are negligible in FBS because of sterile filtration and heat inactivation of the

serum. Unlike the majority of the studies in the field of EVs, here we analyzed untouched, in situ fixed and cultured cells together with their microenvironment. Since centrifugation may disrupt the limiting membrane of amphiectosomes, the in situ fixation made it possible to observe them in their intact form. The culture medium was gently removed by pipetting from above the cells leaving a thin medium layer only (approximately 150 µL of liquid on the cells). Without any further washing, cells were in situ fixed by 4% paraformaldehyde (PFA) in phosphate buffered saline (PBS) for 20 min at room temperature (RT). The released lEVs were either fixed and captured during the release or were preserved on the gelatin/fibronectin surface coating. After fixation, 3×5 min washes with 50 mM glycine in PBS were carried out. In the case of the non-fluorescent HepG2 and HT29 cells, a lactadherin-based plasma membrane staining was performed (*Kovács et al., 2023*; *Vukman et al., 2020*; *Németh et al., 2023*). Lactadherin (Haematologic Technologies) was conjugated to ATTO488 fluorophore (abcam) according to the instructions of the manufacturer. The lactadherin-ATTO488 conjugate was added to the fixed cells in 1:100 dilution in PBS (for 1 hr, RT) before permeabilization. The unbound lactadherin was removed by washing with PBS (three times, 5 min, RT) and post-fixation was carried out by 4% PFA (20 min, RT). PFA was removed by washes with 50 mM glycine in PBS (three times, 5 min, RT). Blocking and permeabilization of the cells were performed by 10% FBS with 0.1% Triton X-100 (Sigma) in PBS (1 hr, RT). In general, primary antibodies were applied in 1:200 dilution O/N at 4°C in the above blocking and permeabilization solution. Excess primary antibodies were eliminated by washing with the blocking and permeabilization solution (three times, 5 min, RT). The secondary antibodies were applied in 1:1000 dilution in 1% FBS in PBS (1 hr, RT). Unbound secondary antibodies were eliminated by washing (1% FBS, in PBS, two times, 5 min; PBS two times, 5 min; water two times, 5 min) and the samples were mounted in ProLong Diamond with DAPI (Invitrogen).

In order to provide evidence for the plasma membrane origin of the outer membrane layer of amphiectosomes, HEK293 were cultured on glass coverslips (VWR). Reaching 60% confluency, the cells were incubated in expansion medium with 5 µg/mL CF488A-conjugated WGA for 30 min at 37°C. After labeling the surface of the plasma membrane by WGA, cells were washed three times by expansion medium and were cultured for an additional 3 hr. Next, they were fixed by 4% PFA (20 min, RT). LC3B and CD63 labeling were performed as described above.

Microscopic slides were examined by Leica SP8 Lightning confocal microscope with adaptive lightning mode using an HC PL APO CS2 ×63/1.40 OIL objective with hybrid detector. Where we showed released MV-lEVs, they were not joined to cells in another detected Z-plane. The applied lookup tables (LUT) were linear during this study. For image analysis and co-localization studies, we applied Leica LASX software using unprocessed raw images. In case of co-localization studies, a 20% threshold and 10% background settings were applied.

## Multi-channel STED super-resolution imaging

Immunofluorescent labeling for multi-channel STED nanoscopy was performed as in the case of confocal microscopy. The primary antibodies used were: LC3B (rabbit) and CD63 (mouse). Abberior Star 635P Goat anti Mouse and Abberior Star 580 Goat anti Rabbit secondary antibodies for STED microscopy have been obtained from Abberior GmbH. Samples were mounted with SlowFade Diamond Antifade Mountant (Thermo). Immunofluorescence was analyzed using an Abberior Instruments Facility Line STED Microscope system built on an Olympus IX83 fully motorized inverted microscope base (Olympus), equipped with a ZDC-830 TrueFocus Z-drift compensator system, an IX3-SSU ultrasonic stage, a QUADScan Beam Scanner scanning head, APD detectors, and an UPLXAPO60XO ×60 oil immersion objective (NA 1.42). We used the 488, 561, and 640 nm solid-state lasers for imaging, and a 775 nm solid-state laser for STED depletion. Image acquisition was performed using the Imspector data acquisition software (version: 16.3.14278-w2129-win64).

## Purification of sEV fraction

sEV fractions for TEM analysis were separated from serum-free conditioned medium of HEK293T-PalmGFP cells by gravity filtration, differential centrifugation, and tangential flow filtration (TFF Easy, HansaBiomed) as described previously (*Németh et al., 2021*).

## Transmission electron microscopy

Adherent cells (HEK293, HEK293T-PalmGFP, HepG2, HT29, and HL1), as well as mouse (C57BL/6) kidney and liver tissues pieces (approximately 1.5 mm × 1.5 mm) were immersed in and fixed by 4% glutaraldehyde (48 hr, 4°C), post-fixed by 1% osmium tetroxide (2 hr, RT) and were embedded into EPON resin (Electron Microscopy Sciences) as described previously (*Olah et al., 1992*). In the case of BMMCs, 920 µL cell suspension was complemented with 80 µL 50% glutaraldehyde to reach the final 4% glutaraldehyde concentration. Cells were fixed for 48 hr at 4°C and were post-fixed by 1% osmium tetroxide (2 hr, RT). During sample preparation, BMMCs were collected by gravity-based sedimentation. Due to the high viscosity of EPON resin, BMMCs were embedded in LR White low viscosity resin (SPI Supplies) according to the instructions of the manufacturer. Ultrathin sections (60 nm) were contrasted by uranyl acetate (3.75%, 10 min, RT) and lead citrate (12 min, RT).

For immunogold labeling of ultrathin sections, cells and tissues were fixed by 4% PFA with 0.1% glutaraldehyde (48 hr, 4°C) and were post-fixed by 0.5% osmium tetroxide (30 min, RT). Samples were embedded into LR White hydrophilic resin. The sections were exposed to $H_2O_2$ and $NaBH_4$ to render the epitopes accessible and were immunogold labeled as described previously (*Valcz et al., 2019*). The contrast was enhanced by uranyl acetate (3.75%, 1 min, RT) and lead citrate (2 min, RT).

HEK293T-PalmGFP-derived sEVs separated from serum-free conditioned medium were detected by negative-positive contrasting without embedding and sectioning (*Théry et al., 2006*). Immunogold labeling was performed as described previously (*Koncz et al., 2023*). Antibodies were used in 1:50 dilution.

A detailed list of the used antibodies is available in the Key resources table.

For all electron microscopic studies, a JEOL 1011 TEM was used. Images were captured with the help of Olympus iTEM software and for image analysis, ImageJ software was used.

## Live-cell imaging

The HEK293T-PalmGFP-LC3RFP stable cell line was cultured the same way as HEK293T-PalmGFP cells. Before the experiments, gelatin-fibronectin-coated 10-well coverslip bottom chamber slide (Greiner-BioOne) was seeded and treated by 30 µM Chloroquine O/N. Release of migrasomes, amphiectosomes, and sEVs were followed by the Leica SP8 Lightning confocal microscope equipped with an Okolab environmental chamber and a Zeiss ELYRA 7 with Lattice SIM² super-resolution fluorescent microscope with the help of ×63/1.4 plan apochromat Oil objective. For image analysis, we applied Leica LASX, Zeiss ZEN Blue, and ImageJ software.

## Modulation of amphiectosome release

To test the release mechanism of amphiectosomes and to distinguish them from migrasomes, different treatments were applied O/N in fresh, serum containing cell culture medium except for Colchicine, where 1 hr treatment was selected. Maturation and fusion of endosomes and lysosomes were inhibited by 30 µM Chloroquine (Invitrogen) or 10 nM BafilomycinA1 (Sigma). Actin polymerization was inhibited by 125 ng/mL Cytochalasin B (Sigma). Tubulin polymerization and function were inhibited by 250 pg/mL Colchicine, while an autophagy-related degradation was induced by 50 ng/mL Rapamycin. The selected concentrations were determined based on both literature data and our preliminary experiments (*Figure 3—figure supplement 1*). Cellular metabolic activity was determined by a metabolic activity-based Resazurin assay (*Koncz et al., 2023*). Fresh cell culture medium was added to control cultures a day before the in situ fixation. Reagents were diluted in fresh cell culture medium. Leica TCS SP8 Lightning confocal microscope was used for detection of amphiectosome release. A few hundred µm² sized area with 15–20 µm in height was tile-scanned with a few hundred cells. The MV-lEVs were recognized as CD63 positive EVs surrounded by GFP positive membrane. They were counted and were normalized to the number of nuclei. Raw images were deposited in BioImage Archive (https://www.ebi.ac.uk/bioimage-archive/) with the accession number S-BIAD1456 (doi: https://www.ebi.ac.uk/biostudies/studies/S-BIAD1456).

## Western blotting

Presence of proteins and specificity of the used primary antibodies were confirmed by western blotting as described previously (*Koncz et al., 2023*). For accurate quantification (free from variations potentially caused by EV purification), we analyzed cell-, serum- and lEV (diameter>800 nm) free

conditioned medium. The cells were cultured O/N in a serum-free culture medium. After harvesting cells were eliminated by centrifugation (300×$g$, 10 min at 4°C) followed by a 2000×$g$ centrifugation (30 min at 4°C) to eliminate lEVs. Total protein content of the conditioned, serum-, cell- and lEV-free medium was precipitated by trichloroacetic acid as described previously (*Koncz et al., 2023*; *Koontz, 2014*). The protein pellets were suspended in cOmplete Protease Inhibitor Cocktail (Roche) containing radio-immunoprecipitation assay (RIPA) buffer.

When whole-cell lysate was tested for validation of antibodies and the HEK293T-PalmGFP-LC3RFP cell line, cells were lysed in cOmplete Protease Inhibitor Cocktail (Roche) containing RIPA buffer.

Polyacrylamide gel electrophoresis was carried out using 10% gels (acrylamide/bis-acrylamide ratio 37.5:1) or any kDa precast gels (Bio-Rad) and a MiniProtean (Bio-Rad) gel running system. For better solubilization of membrane proteins, equal volumes of 0.1% Triton X-100, Laemmli buffer, and samples were mixed as described previously (*Visnovitz et al., 2012*). Approximately 10–30 µg protein were loaded into each well. Following electrophoretic separation, proteins were transferred to PVDF membranes (Serva). Membranes were blocked with 5% skimmed milk powder or 5% BSA in washing buffer for 1 hr. Primary antibodies were applied in 1:1000 dilution except for the anti-CD63 (rabbit), anti-CD81 (rabbit), and anti-CD81 (mouse) antibodies where 1:500, 1:2500, and 1:100 dilutions were used, respectively. Peroxidase-labeled secondary antibodies were applied in 1:10,000 dilution. The signals were detected by ECL Western Blotting Substrate (Thermo Scientific) with an Imager CHEMI Premium (VWR) image analyzer system. In case of quantification, equal protein amounts were loaded to the gels. Within a biological replicate, the control and Chloroquine-treated samples were run on the same gels. To enable comparison, the relative expression of control and Chloroquine-treated samples were determined and compared.

## Software and statistical analysis

For image capturing, analysis, and co-localization studies, Leica LAS X, Zeiss ZEN Blue, Olympus iTEM, and ImageJ software were used. Figures and graphs were generated using GraphPad Prism 9.4.1 and BioRender (BioRender.com). For statistical analysis, standard deviation was calculated. Unpaired two-tailed Student's t-tests and one-way ANOVA were used (*$p<0.05$, **$p<0.01$, ***$p<0.001$, ****$p<0.0001$).

## Acknowledgements

This research was funded by the NVKP_16-1-2016-0004 grant of the Hungarian National Research, Development and Innovation Office (NKFIH), ÚNKP-23-3-I-SE-2, the Semmelweis Innovation Fund (STIA 2020 KFI), Hungarian Scientific Research Fund (OTKA K120237, K135637, Advanced 150767, and FK 138851), Eötvös Loránd University Excellence Fund (EKA 2022/045-P101), Hungary Academy of Sciences, LP2022-13/2022, VEKOP-2.3.2-162016-00002, VEKOP-2.3.3-15-2017-00016, the Higher Education Excellence Program (FIKP) and the Therapeutic Thematic Programme TKP2021-EGA-23. This study was also supported by the grants RRF-2.3.121-2022-00003 (National Cardiovascular Laboratory Program) and 2019-2.1.7-ERA-NET-2021-00015. The project has received funding from the EU's Horizon 2020 Research and Innovation Programme under grant agreement No. 739593, 'Momentum' research grant from the Hungarian Academy of Sciences (LP2022-5/2022) and the Hungarian Brain Research Program NAP2022-I-1/2022. TV and CC were supported by the János Bolyai Research Scholarship of the Hungarian Academy of Sciences. The authors would like to thank the ZEISS Microscopy Customer Center Team for the collaboration, and Dr. Abel Pereira da Graça for the possibility to use the Elyra7 SIM (*György et al., 2011*) super-resolution microscope. The authors are grateful to Györgyné Vidra, Györgyi Balogh, and Andrea Orbán for their technical help and advises.

## Additional information

### Competing interests

Gábor Valcz: Employee of 3DHISTECH Ltd. Edit I Buzas: EIB is a member of the Advisory Board of Sphere Gene Therapeutics Inc (Boston, MA, USA) and ReNeuron (UK). The other authors declare that no competing interests exist.

## Funding

| Funder | Grant reference number | Author |
| --- | --- | --- |
| NKFIH | NVKP_16-1-2016-0004 | Edit I Buzas |
| NKFIH | ÚNKP-23-3-I-SE-2 | Dorina Lenzinger |
| Semmelweis Innovation Fund | STIA 2020 KFI | Tamás Visnovitz |
| NKFIH | K120237 | Edit I Buzas |
| NKFIH | K135637 | Zsolt I Komlósi |
| NKFIH | Advanced 150767 | Edit I Buzas |
| NKFIH | FK 138851 | Péter Lőrincz |
| Eotvos Lorand University Excellence Fund | EKA 2022/045-P101 | Péter Lőrincz |
| Hungary Academy of Sciences | LP2022-13/2022 | Péter Lőrincz |
| NKFIH | Therapeutic Thematic Programme TKP2021-EGA-23 | Edit I Buzas |
| Horizon 2020 Framework Programme | 739593 | Edit I Buzas |
| Hungarian Academy of Sciences | LP2022-5/2022 | Ádám Dénes |
| Hungarian Academy of Sciences | János Bolyai Research Scholarship | Tamás Visnovitz Csaba Cserép |
| Hungarian Government and European Commission | VEKOP-2.3.2-162016-00002 | Edit I Buzas |
| Hungarian Government and European Commission | VEKOP-2.3.3-15-2017-00016 | Edit I Buzas |
| NKFIH | Higher Education Excellence Program (FIKP) | Edit I Buzas |
| NKFIH | RRF-2.3.121-2022-00003 | Edit I Buzas |
| European Commission | 2019-2.1.7-ERA-NET-2021-00015 | Edit I Buzas |
| NKFIH | Hungarian Brain Research Program NAP2022-I-1/2022 | Ádám Dénes |

The funders had no role in study design, data collection and interpretation, or the decision to submit the work for publication.

## Author contributions

Tamás Visnovitz, Conceptualization, Formal analysis, Supervision, Funding acquisition, Investigation, Methodology, Writing – original draft; Dorina Lenzinger, Formal analysis, Funding acquisition, Investigation; Anna Koncz, Formal analysis, Investigation; Péter M Vizi, Tünde Bárkai, Alicia Galinsoga, Kelsey Fletcher, Investigation; Krisztina V Vukman, Methodology; Krisztina Németh, Formal analysis; Zsolt I Komlósi, Csaba Cserép, Funding acquisition, Investigation, Methodology; Ádám Dénes, Resources, Funding acquisition; Péter Lőrincz, Funding acquisition, Methodology; Gábor Valcz, Conceptualization, Writing – original draft; Edit I Buzas, Conceptualization, Resources, Supervision, Funding acquisition, Writing – original draft

## Author ORCIDs

Tamás Visnovitz (ID) http://orcid.org/0000-0002-7962-5083
Dorina Lenzinger (ID) https://orcid.org/0000-0003-0270-7985
Péter Lőrincz (ID) https://orcid.org/0000-0001-7374-667X
Edit I Buzas (ID) https://orcid.org/0000-0002-3744-206X

Reviewer #1 (Public review): https://doi.org/10.7554/eLife.95828.3.sa1
Reviewer #2 (Public review): https://doi.org/10.7554/eLife.95828.3.sa2
Author response https://doi.org/10.7554/eLife.95828.3.sa3

## Additional files

### Supplementary files
MDAR checklist

### Data availability
Full Western blot images are included in Figure 2—figure supplement 4—source data 1-2, Figure 2—figure supplement 5—source data 1-2 and Figure 3—source data 2-3. Raw images of Figure 3I were deposited in BioImage Archive (https://www.ebi.ac.uk/bioimage-archive/) with the accession number S-BIAD1456 (doi: 10.6019/S-BIAD1456). Source data of the graphs in figures have been provided in source data files (Figure 2-source data 1, Figure 2-figure supplement 1-source data 1, Figure 2-figure supplement 2-source data 1, Figure 2-figure supplement 6-source data 1, Figure 3-source data 1, Figure 3-figure supplement 1-source data, Figure 3-source data 1).

The following dataset was generated:

| Author(s) | Year | Dataset title | Dataset URL | Database and Identifier |
|---|---|---|---|---|
| Visnovitz T, Buzás EI | 2024 | Original images for A "torn bag mechanism" of small extracellular vesicle release via limiting membrane rupture of en bloc released amphisomes (amphiectosomes) | https://doi.org/10.6019/S-BIAD1456 | BioImages Archive, 10.6019/S-BIAD1456 |

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
