## [Editor Report · eLife Assessment]

In this study, the authors present **compelling** data illustrating a potential mechanism for a hitherto not described form of extracellular vesicle biogenesis. Their model suggests that small extracellular vesicles are secreted from cells within larger vesicles, termed amphiectosomes, which subsequently rupture to release their smaller vesicle contents. This discovery represents an **important** advancement in the field.

---

## [Referee Report · Reviewer #1 (Public review)]

Summary:

The authors' research group had previously demonstrated the release of large multivesicular body-like structures by human colorectal cancer cells. This manuscript expands on their findings, revealing that this phenomenon is not exclusive to colorectal cancer cells but is also observed in various other cell types, including different cultured cell lines, as well as cells in the mouse kidney and liver. Furthermore, the authors argue that these large multivesicular body-like structures originate from intracellular amphisomes, which they term "amphiectosomes." These amphiectosomes release their intraluminal vesicles (ILVs) through a "torn-bag mechanism." Finally, the authors demonstrate that the ILVs of amphiectosomes are either LC3B positive or CD63 positive. This distinction implicates that the ILVs either originate from amphisomes or multivesicular bodies, respectively.

Strengths:

The manuscript reports a potential origin of extracellular vesicle (EV) biogenesis. The reported observations are intriguing.

Weaknesses:

In their revised version, the authors have addressed the majority of my criticisms. I have no further concerns regarding this manuscript.

---

## [Referee Report · Reviewer #2 (Public review)]

Summary:

authors had previously identified that a colorectal cancer cell line generates small extracellular vesicles (sEVs) via a mechanism where a larger intracellular compartment containing these sEVs is secreted from the surface of the cell and then tears to release its contents. Previous studies had suggested that intraluminal vesicles (ILVs) inside endosomal multivesicular bodies and amphisomes can be secreted by fusion of the compartment with the plasma membrane. The 'torn bag mechanism' considered in this manuscript is distinctly different, because it involves initial budding off of a plasma membrane-enclosed compartment (called the amphiectosome in this manuscript, or MV-lEV). The authors successfully set out to investigate whether this mechanism is common to many cell types and to determine some of the subcellular processes involved.

The strengths of the study are:

(1) The high-quality imaging approaches used, including live-cell imaging and EN, which seem to show good examples of the proposed mechanism.

(2) They screen several cell lines for these structures, also search for similar structures in vivo, and show the tearing process by real-time imaging.

(3) Regarding the intracellular mechanisms of ILV production, the authors also try to demonstrate the different stages of amphiectosome production and differently labelled ILVs using immuno-EM.

Several of the techniques employed are technically challenging to do well, and so these are critical strengths of the manuscript.

Overall, I think the authors have been successful in identifying amphiectosomes secreted from multiple cell lines and cells in vivo, and in demonstrating that the ILVs inside them have at least two origins (autophagosome membrane and late endosomal multivesicular body) based on the markers that they carry. Inevitably, it remains unclear how universal this mechanism is in vivo and its overall contribution to EV function.

I think there could be a significant impact on the EV field and consequently on our understanding of cell-cell signalling based on these findings. It will flag the importance of investigating the release of amphiectosomes in other studies, especially as the molecular mechanisms involved in this type of 'ectosomal-style' release will be different from multivesicular compartment fusion to the plasma membrane and should be possible to be manipulated independently.

In general, the EV field has struggled to link up analysis of the subcellular biology of sEV secretion and the biochemical/physical analysis of the sEVs themselves, so from that perspective, the manuscript provides a novel angle on this problem.

---

## [Author Response]

The following is the authors’ response to the original reviews.

**Public Reviews:**

**Reviewer #1 (Public Review):**
Summary:The authors' research group had previously demonstrated the release of large multivesicular body-like structures by human colorectal cancer cells. This manuscript expands on their findings, revealing that this phenomenon is not exclusive to colorectal cancer cells but is also observed in various other cell types, including different cultured cell lines, as well as cells in the mouse kidney and liver. Furthermore, the authors argue that these large multivesicular body-like structures originate from intracellular amphisomes, which they term "amphiectosomes." These amphiectosomes release their intraluminal vesicles (ILVs) through a "torn-bag mechanism." Finally, the authors demonstrate that the ILVs of amphiectosomes are either LC3B positive or CD63 positive. This distinction implies that the ILVs either originate from amphisomes or multivesicular bodies, respectively.Strengths:The manuscript reports a potential origin of extracellular vesicle (EV) biogenesis. The reported observations are intriguing.Weaknesses:It is essential to note that the manuscript has issues with experimental designs and lacks consistency in the presented data. Here is a list of the major concerns:(1) The authors culture the cells in the presence of fetal bovine serum (FBS) in the culture medium. Given that FBS contains a substantial amount of EVs, this raises a significant issue, as it becomes challenging to differentiate between EVs derived from FBS and those released by the cells. This concern extends to all transmission electron microscopy (TEM) images (Figure 1, 2P-S, S5, Figure 4 P-U) and the quantification of EV numbers in Figure 3. The authors need to use an FBS-free cell culture medium.

Although FBS indeed contains bovine EVs, however, the presence of very large multivesicular EVs (amphiectosomes) that our manuscript focuses on has never been observed and reported. For reported size distributions of EVs in FBS, please find a few relevant references below:

PMID: 29410778, PMID: 33532042, PMID: 30940830 and PMID: 37298194

All the above publications show that the number of lEVs > 350-500 nm is negligible in FBS. The average diameter of MV-lEVs (amphiectosomes) described in our manuscript is around 1.00-1.50 micrometer.

**Reviewer #1:** These papers evaluated the effectiveness of various methods to eliminate EVs from FBS, emphasizing the challenges associated with the presence of EVs in FBS. They also caution against using FBS in EV studies due to these issues. However, I did not find a clear indication regarding the size distributions of EVs in FBS in these papers.Please provide accurate reference supporting the claim that 'lEVs > 350-500 nm are negligible in FBS.' The papers cited by the authors do not address this specific point.

In the revised manuscript, we addressed the point that due to sterile filtering of FBS, it cannot contain large >0.22 µm EVs

**Our response to Reviewer #1 point 2.** When we demonstrated the TEM of isolated EVs, we consistently used serum- free conditioned medium (Fig2 P-S, Fig2S5 J, O) as described previously (Németh et al 2021, PMID: 34665280).

**Reviewer #1**: This is an important point that is not mentioned in the original main text, figure legend or method. Please address.

We agree and we apologize for it. We added this information to the revised manuscript.

**Our response to Reviewer #1 point 3**. Our TEM images show cells captured in the process of budding and scission of large multivesicular EVs excluding the possibility that these structures could have originated from FBS.

**Reviewer #1:** These images may also depict the engulfment of EVs in FBS. Hence, it is crucial to utilize EV-free or EV-depleted FBS.

As we mentioned earlier, we added the information to the revised manuscript that sterile filtering of the FBS presumably removed particles >0.22 µm EVs

**Our response to Reviewer #1 point** 4. In addition, in our confocal analysis, we studied Palm-GFP positive, cell-line derived MV-lEVs. Importantly, in these experiments, FBS-derived EVs are non-fluorescent, therefore, the distinction between GFP positive MV-lEVs and FBS-derived EVs was evident.

**Reviewer #1:** I agree that these fluorescent-labeled assays conclusively indicate that the MV-lEVs are originating from the cells. However, the images of concerns are the non- fluorescent-labeled images in (Figure 1, 2P-S, S5, Figure 4 P-U and Figure 3). The MV-lEVs may derive from both the cells and FBS.

Please see above our response to points 1-3.

**Our response to Reviewer #1 point5**. In addition, culturing cells in FBS-free medium (serum starvation) significantly affects autophagy. Given that in our study, we focused on autophagy related amphiectosome secretion, we intentionally chose to use FBS supplemented medium.

**Reviewer #1** If this is a concern, the authors should use EV-depletive FBS.

As we discussed above, sterile filtration of FBS removes particles >0.22 µm. In addition, based on our preliminary experiments, EV-depleted serum may effect cell physiology.

**Our response to Reviewer #1 point6**. Even though the authors of this manuscript are not familiar with the technological details how FBS is processed before commercialization, it is reasonable to assume that the samples are subjected to sterile filtration (through a 0.22 micron filter) after which MV-lEVs cannot be present in the commercial FBS samples.

**Reviewer #1**This is a fair comment that needs to be included in the manuscript.

As you suggested, this comment is now included in the revised manuscript

(2) The data presented in Figure 2 is not convincingly supportive of the authors' conclusion. The authors argue that "...CD81 was present in the plasma membrane-derived limiting membrane (Figures 2B, D, F), while CD63 was only found inside the MV-lEVs (Fig. 2A, C, E)." However, in Figure 2G, there is an observable CD63 signal in the limiting membrane (overlapping with the green signals), and in Figure 2J, CD81 also exhibits overlap with MV-IEVs.

Both CD63 and CD81 are tetraspanins known to be present both in the membrane of sEVs and in the plasma membrane of cells (for references, please see Uniprot subcellular location maps: https://www.uniprot.org/uniprotkb/P08962/entry#subcellular_location
https://www.uniprot.org/uniprotkb/P60033/entry#subcellular_location). However, according the feedback of the reviewer, for clarity, we will delete the implicated sentence from the text.

**Reviewer #1** Please also justify the statement questioned in (3) as these arguments are interconnected.

We hope you find our above responses to your comment acceptable.

(3) Following up on the previous concern, the authors argue that CD81 and CD63 are exclusively located on the limiting membrane and MV-IEVs, respectively (Figure 2-A-M). However, in lines 104-106, the authors conclude that "The simultaneous presence of CD63, CD81, TSG101, ALIX, and the autophagosome marker LC3B within the MV-lEVs..." This statement indicates that CD63 and CD81 co-localize to the MV-IEVs. The authors need to address this apparent discrepancy and provide an explanation.

There must be a misunderstanding because we did not claim or implicate in the text that “CD81 and CD63 are exclusively located on the limiting membrane and MV-IEVs”. Here we studied co-localization of the above proteins in the case intraluminal vesicles (ILVs). In Fig 2. we did not show any analysis of limiting membrane co-localization.

**Reviewer #1** I have indicated that this statement is found in lines 104-106, where the authors argue, 'The simultaneous presence of CD63, CD81, TSG101, ALIX, and the autophagosome marker LC3B within the MV-lEVs...' If the authors acknowledge the inaccuracy of this statement, please provide a justification for this argument.

For clarity, we modified the description of data shown in Fig2 in the revised manuscript.

(4) The specificity of the antibodies used in Figure 2 should be validated through knockout or knockdown experiments. Several of the antibodies used in this figure detect multiple bands on western blots, raising doubts about their specificity. Verification through additional experimental approaches is essential to ensure the reliability and accuracy of all the immunostaining data in this manuscript.

We will consider this suggestion during the revision of the manuscript.

**Reviewer #1:**Please do so.

We carefully considered the suggestion, but we realized that it was not feasible for us to perform gene silencing in the case of all our used antibodies before resubmission of our revised manuscript. However, we repeated the Western blot for mouse anti-CD81 (Invitrogen MAA5-13548) and replaced the previous Western blot by it in the revised manuscript (Fig.2-S4H)

(5) In Figures 2P-R, the morphology of the MV-IEVs does not resemble those shown in Figures 1-A, H, and D, indicating a notable inconsistency in the data.

EM images in Figure2 P-R show sEVs separated from serum-free conditioned media as opposed to MV-lEVs, which were in situ captured in fixed tissue cultures (Fig1). Therefore, the two EV populations necessarily have different size and structure. Furthermore, Fig. 1 shows images of ultrathin sections while in Figure 2P-R, we used a negative-positive contrasting of intact sEV-s without embedding and sectioning.

(6) There are no loading controls provided for any of the western blot data.

Not even the latest MISEV 2023 guidelines give recommendations for proper loading control for separated EVs in Western blot (MISEV 2023 , DOI: 10.1002/jev2.12404 PMID: 38326288). Here we applied our previously developed method (PMID: 37103858), which in our opinion, is the most reliable approach to be used for sEV Western blotting. For whole cell lysates, we used actin as loading control (Fig3-S2B).

**Reviewer #1:** The blots referenced here (Fig2-S3; Fig2-S4B; Fig3-S2B) were conducted using total cell lysates, not EV extracts. Only one blot in Fig3-S2B includes an actin control. All remaining blots should incorporate actin controls for consistency.

Fig2-S3 (corresponding to Fig2-S4 in the revised manuscript) only shows reactivity of the used antibodies. This Western blot is not intended to serve as a basis of any quantitative conclusions. Fig2-S4 (corresponding to Fig2-S5 in the revised manuscript) includes the actin control. Fig3-S2B shows the complete membrane, which was cut into 4 pieces, and the immune reactivity of different antibodies was tested. The actin band was included on the anti-LC3B blot. For clarity, we rephrased the figure legend.

Additionally, for Figures 2-S4B, the authors should run the samples from lanes i-iii in a single gel.

Please note that in Figure 2- S4B, we did run a single gel, and the blot was cut into 4 pieces, which were tested by anti-GFP, anti-RFP, anti-LC3A and anti-LC3B antibodies. Full Western blots are shown in Fig.3_S2 B, and lanes “1”, “2” and “3” correspond to “i”, “ii” and “iii” in Fig.2-S4, respectively.

**Reviewer #1:** In the original Figure 2- S4B, the blots were sectioned into 12 pieces. If lanes "i," "ii," and "iii" were run on the same blot, the authors are advised to eliminate the grids between these lanes.

Grids separating the lanes have been eliminated on Fig.2_S4 (now Fig.2_S5 in the revised manuscript).

(7) In Figure 2-S4, is there co-localization observed between LC3RFP (LC3A?) with other MV-IFV markers? How about LC3B? Does LC3B co-localize with other MV-IFV markers?

In Supplementary Figure 2-S4, we showed successful generation of HEK293T-PalmGFP-LC3RFP cell line. In this case we tested the cells, and not the released MV-lEVs. LC3A co-localized with the RFP signal as expected.

**Reviewer #1:** Does LC3RFP colocalize with MV-IFV markers in HEK293T-PalmGFP-LC3RFP cell line? This experiment aims to clarify the conclusion made in lines 104-106, where the authors assert that 'The concurrent existence of CD63, CD81, TSG101, ALIX, and the autophagosome marker LC3B within the MV-lEVs...'

In the case of PalmGFP-LC3RFP cells, LC3-RFP is overexpressed. Simultaneous assessment of this overexpressed protein with non-overexpressed, fluorescent antibod-detected molecules proved to be challenging because of spectral overlaps and inappropriate signal-noise ratios. Furthermore, in association with EVs, the number of antibody-detected molecules is substantially lower than in cells. Therefore, even though we tried, we could not successfully perform these experiments.

(8) The TEM images presented in Figure 2-S5, specifically F, G, H, and I, do not closely resemble the images in Figure 2-S5 K, L, M, N, and O. Despite this dissimilarity, the authors argue that these images depict the same structures. The authors should provide an explanation for this observed discrepancy to ensure clarity and consistency in the interpretation of the presented data.

As indicated in Material and Methods, Fig 2-S5 F, G, H and I are conventional TEM images fixed by 4% glutaraldehyde 1% OsO_4_ 2h and embedded into Epon resin with a post contrasting of 3.75% uranyl acetate 10 min and 12 min lead citrate. Samples processed this way have very high structure preservation and better image quality, however, they are not suitable for immune detection. In contrast, Fig.2.-S5 K,L,M,N shows immunogold labelling of in situ fixed samples. In this case we used milder fixation (4% PFA, 0.1% glutaraldehyde, postfixed by 0.5% OsO_4_ 30 min) and LR-White hydrophilic resin embedding. This special resin enables immunogold TEM analysis. The sections were exposed to H_2_O_2_ and NaBH_4_ to render the epitopes accessible in the resin. Because of the different applied techniques, the preservation of the structure is not the same. In the case of Fig.2 J, O, separated sEVs were visualised by negative-positive contrast and immunogold labelling as described previously (PMID: 37103858).

**Reviewer #1:** Please include this justification in the revised version.

We included this justification in the revised manuscript.

(9) For Figures 3C and 3-S1, the authors should include the images used for EV quantification. Considering the concern regarding potential contamination introduced by FBS (concern 1), it is advisable for the authors to employ an independent method to identify EVs, thereby confirming the reliability of the data presented in these figures.

In our revised manuscript, we will provide all the images used for EV quantification in Figure 3C. Given that Figures 3C and 3-S1 show MV-lEVs released by HEK293T-PlamGFP cells, the possible interference by FBS-derived non-fluorescent EVs can be excluded.

**Reviewer #1:** Please provide all the images.

Original LASX files are provided (DOI: 10.6019/S-BIAD1456).

**Reviewer #1:** The images raising concerns regarding the contamination of EVs in FBS primarily consist of transmission electron microscopy (TEM) images, namely, Figure 1, 2P-S, S5, and Figure 4 P-U, along with the quantification of EV numbers in Figure 3. These concerns persist despite the use of fluorescent-labeled experiments. While fluorescent-labeled MV-lEVs are conclusively identified as originating from the cells, the MV-lEVs observed in Figure 1, 2P-S, S5, and Figure 4 P-U and Figure 3 may derive from both the cells and FBS.

Large EVs (with diameter >800 nm) derived from FBS were not present in our experiments, as discussed above.

(10) Do the amphiectosomes released from other cell types as well as cells in mouse kidneys or liver contain LC3B positive and CD63 positive ILVs?

Based on our confocal microscopic analysis, in addition the HEK293T-PalmGFP cells, HT29 and HepG2 cells also release similar LC3B and CD63 positive MV-lEVs. Preliminary evidence shows MV-lEV secretion by additional cell types.

**The response of Reviewer #1:** Please show these data in the revised manuscript. Moreover, do cells in mouse kidneys or liver contain LC3B positive and CD63 positive ILVs?

We have added new confocal microscopic images to Fig2-S3 showing amphiectosomes released also by the H9c2 (ATCC) cardiomyoblast cell line. To preserve the ultrastructure of MV-lEVs in complex organs like kidney and liver, fixation with 4% glutaraldehyde with 1% OsO4 appears to be essential. This fixation does not allow for immune detection to assess LC3B and CD63 positive MV-lEVs in the ultrathin sections.

**Reviewer #2 (Public Review):**
Summary:The authors had previously identified that a colorectal cancer cell line generates small extracellular vesicles (sEVs) via a mechanism where a larger intracellular compartment containing these sEVs is secreted from the surface of the cell and then tears to release its contents. Previous studies have suggested that intraluminal vesicles (ILVs) inside endosomal multivesicular bodies and amphisomes can be secreted by the fusion of the compartment with the plasma membrane. The 'torn bag mechanism' considered in this manuscript is distinctly different because it involves initial budding off of a plasma membrane-enclosed compartment (called the amphiectosome in this manuscript, or MV-lEV). The authors successfully set out to investigate whether this mechanism is common to many cell types and to determine some of the subcellular processes involved.The strengths of the study are:(1) The high-quality imaging approaches used, seem to show good examples of the proposed mechanism.(2) They screen several cell lines for these structures, also search for similar structures in vivo, and show the tearing process by real-time imaging.(3) Regarding the intracellular mechanisms of ILV production, the authors also try to demonstrate the different stages of amphiectosome production and differently labelled ILVs using immuno-EM.Several of these techniques are technically challenging to do well, and so these are critical strengths of the manuscript.The weaknesses are:(1) Most of the analysis is undertaken with cell lines. In fact, all of the analysis involving the assessment of specific proteins associated with amphiectosomes and ILVs are performed in vitro, so it is unclear whether these processes are really mirrored in vivo. The images shown in vivo only demonstrate putative amphiectosomes in the circulation, which is perhaps surprising if they normally have a short half-life and would need to pass through an endothelium to reach the vessel lumen unless they were secreted by the endothelial cells themselves.

Our previous results analyzing PFA-fixed, paraffin embedded sections of colorectal cancer patients provided direct evidence that MV-lEV secretion also occurs in humans in vivo (PMID: 31007874). Regarding your comment on the presence of amphiectosomes in the circulation despite their short half-lives, we would like to point out that Fig1.X shows a circulating lymphocyte which releases MV-lEV within the vessel lumen. Furthermore, in the revised manuscript, an additional Fig.1-S1 is provided. Here, we show the release of MV-lEVs both by an endothelial and a sub-endothelial cell (Fig.1-S1G). In addition, these images show the simultaneous presence of MV-lEVs and sEVs in the circulation (Fig.1-S1.A,C,D,H and I). The transmission electron micrographs of mouse kidney and liver sections provide additional evidence that the MV-lEVs are released by different types of cells, and the “torn bag release” also takes place in vivo (Fig.1.V).

(2) The analysis of the intracellular formation of compartments involved in the secretion process (Figure 2-S5) relies on immuno-EM, which is generally less convincing than high-/super-resolution fluorescence microscopy because the immuno-labelling is inevitably very sporadic and patchy. High-quality EM is challenging for many labs (and seems to be done very well here), but high-/super-resolution fluorescence microscopy techniques are more commonly employed, and the study already shows that these techniques should be applicable to studying the intracellular trafficking processes.

As you suggested, in the revised manuscript, we present additional super-resolution microscopy (STED) data. The intracellular formation of amphisomes, the fragmentation of LC3B-positive membranes and the formation of LC3B-positive ILVs were captured (Fig. 3B-F).

(3) One aspect of the mechanism, which needs some consideration, is what happens to the amphisome membrane, once it has budded off inside the amphiectosome. In the fluorescence images, it seems to be disrupted, but presumably, this must happen after separation from the cell to avoid the release of ILVs inside the cell. There is an additional part of Figure 1 (Figure 1Y onwards), which does not seem to be discussed in the text (and should be), that alludes to amphiectosomes often having a double membrane.

We agree with your comment regarding the amphisome membrane and we added a sentence to the Discussion of the revised manuscript. Fig1Y onwards is now discussed in the manuscript. In addition, we labelled the surface of living HEK293 cells with wheat germ agglutinin (WGA), which binds to sialic acid and N-acetyl-D-glucosamine. After removing the unbound WGA by washes, the cells were cultured for an additional 3 hours, and the release of amphiectosomes was studied. The budding amphiectosome had WGA positive membrane providing evidence that the external limiting membrane had a plasma membrane origin (Fig.3G)

(4) The real-time analysis of the amphiectosome tearing mechanism seemed relatively slow to me (over three minutes), and if this has been observed multiple times, it would be helpful to know if this is typical or whether there is considerable variation.

Thank you for this comment. In the revised manuscript, we highlight that the first released LC3 positive ILV was detected as early as within 40 sec.

Overall, I think the authors have been successful in identifying amphiectosomes secreted from multiple cell lines and demonstrating that the ILVs inside them have at least two origins (autophagosome membrane and late endosomal multivesicular body) based on the markers that they carry. The analysis of intracellular compartments producing these structures is rather less convincing and it remains unclear what cells release these structures in vivo.I think there could be a significant impact on the EV field and consequently on our understanding of cell-cell signalling based on these findings. It will flag the importance of investigating the release of amphiectosomes in other studies, and although the authors do not discuss it, the molecular mechanisms involved in this type of 'ectosomal-style' release will be different from multivesicular compartment fusion to the plasma membrane and should be possible to be manipulated independently. Any experiments that demonstrate this would greatly strengthen the manuscript.

We appreciate these comments of the reviewer. Experiments are on their way to elucidate the mechanism of the “ectosomal style” exosome release and will be the topic of our next publication.

In general, the EV field has struggled to link up analysis of the subcellular biology of sEV secretion and the biochemical/physical analysis of the sEVs themselves, so from that perspective, the manuscript provides a novel angle on this problem.
**Reviewer #3 (Public Review):**
Summary:In this manuscript, the authors describe a novel mode of release of small extracellular vesicles. These small EVs are released via the rupture of the membrane of so-called amphiectosomes that resemble "morphologically" Multivesicular Bodies.These structures have been initially described by the authors as released by colorectal cancer cells (https://doi.org/10.1080/20013078.2019.1596668). In this manuscript, they provide experiments that allow us to generalize this process to other cells. In brief, amphiectosomes are likely released by ectocytosis of amphisomes that are formed by the fusion of multivesicular endosomes with autophagosomes. The authors propose that their model puts forward the hypothesis that LC3 positive vesicles are formed by "curling" of the autophagosomal membrane which then gives rise to an organelle where both CD63 and LC3 positive small EVs co-exist and would be released then by a budding mechanism at the cell surface that appears similar to the budding of microvesicles /ectosomes. Very correctly the authors make the distinction from migrasomes because these structures appear very similar in morphology.Strengths:The findings are interesting despite that it is unclear what would be the functional relevance of such a process and even how it could be induced. It points to a novel mode of release of extracellular vesicles.Weaknesses:This reviewer has comments and concerns concerning the interpretation of the data and the proposed model. In addition, in my opinion, some of the results in particular micrographs and immunoblots (even shown as supplementary data) are not of quality to support the conclusions.
**Recommendations for the authors:**

**Reviewer #1 (Recommendations For The Authors):**
(1) Highlight MV-IEV, ILV and limiting membrane in Figure-1G, N, and U.

Based on the suggestion, we revised Figure1

(2) Figure 1-Y-AF are not mentioned in the text.

In the revised manuscript, we discuss Figure 1Y-AF

(3) The term "IEVs" in Figure 2-S2 is not defined.

We modified the figure legend: we changed MV-lEV to amphiectosome

(4) Need to quantify co-localization in Figure 2-S2.

As suggested, we carried out the co-localisation analysis (Fig2-S2I), and Fig2-S2 was re-edited

**Reviewer #2 (Recommendations For The Authors):**
I have two recommendations for improving the manuscript through additional experiments:(1) I think the description of the intracellular processes taking place in order to form amphiectosomes would be much stronger if some super-resolution imaging could be undertaken. This should label the different compartments before and after fusion with specific markers that highlight the protein signature of the different limiting and ILV membranes much more clearly than immuno-EM. It will also help in characterising the double-membrane structure of amphiectosomes at the point of budding and reveal whether the patchy labelling of the inner membrane emerges after amphiectosome release (the schematic model currently suggests that it happens before).

Thank you for your suggestion. STED microscopy was applied and results are shown in new Fig3 and the schematic model was modified accordingly.

(2) The implications of the manuscript would be more wide-ranging if the authors could test genetic manipulations that are believed to block exosome or ectosome release, eg. Rab27a or Arrdc1 knockdown. This may allow them to determine whether MV-lEVs can be released independently of the classical exosome release mechanism because they use a different route to be released from the plasma membrane. This experiment is not essential, but I think it would start to address the core regulatory mechanisms involved, and if successful, would easily allow the authors to determine the ratio of CD63-positive sEVs being secreted via classical versus amphiectosome routes.

The suggestion is very valuable for us and these studies are being performed in a separate project.

I think there are several other ways in which the manuscript could be improved to better explain some of the approaches, findings and interpretation:

(1) Include some explanation in the text of certain key tools, particularly:a. Palm-GFP and whether its expression might alter the properties of the plasma membrane since this is used in a lot of experiments and is the only marker that seems to uniformly label the outer membrane of amphiectosomes. One concern might be that its expression drives amphiectosome secretion.

We found evidence for amphiectosome release also in the case of several different cells not expressing Palm-GFP. We believe, this excludes the possibility that Palm-GFP expression is the inducer of the amphiectosome release. Both by fluorescent and electron microscopy, the Palm-GFP non expressing cells showed very similar MV-lEVs. In addition, in the case of non-transduced HEK293 and fluorescent WGA-binding, we made similar observations.

b. Lactadherin - does this label the amphiectosomes after their release or does the wash-off step mean that it only labels cells, which subsequently release amphiectosomes?

Lactadherin labels the amphiectosomes after their release and fixation. Living cells cannot be labelled by lactadherin as PS is absent in the external plasma membrane layer of living cells. We used WGA on HEK293 cells to further support the plasma membrane origin of the external membrane of amphiectosomes.

(2) Explain the EM and confocal imaging approaches more clearly. Most importantly, is a 3D reconstruction always involved to confirm that 'separated' amphiectosomes are not joined to cells in another Z-plane.

Thank you for your suggestion. We have modified the manuscript accordingly

(3) Presenting triple-labelled images with red, green and yellow channels does not allow individual labelling to be determined without single-channel images and even then, it is much more informative to use three distinguishable colours that make a different colour with overlap, eg. CMY? Fig.2_S2D and E do not display individual channels, so definitely need to be changed.

In case of Fig.2_S2D, we now show the individual channels, the earlier E image has been removed. In case of the STED images, CMY colors had been used, as you suggested.

(4) Please discuss in the text the data in Figure 1Y onwards concerning single/double membranes on MV-lEVs.

In the revised manuscript, we discuss the question on single/double membranes and we refer to Figure 1Y-AF

(5) On line 162, reword 'intraluminal TSPAN4 only' to 'one in which TSPAN4 is only intraluminal' to make it clear that other proteins are also marking the intraluminal region, not TSPAN4 only.

We modified the text accordingly.

(6) Points for further discussion and further conclusions:a. In vivo experiments - discuss the limitations of this part of the analysis - it seems that none of the amphiectosome markers have been analysed in this part of the study and the MV-lEVs are only in the circulation.b. Can the authors give any further indication of the levels of MV-lEVs relative to free sEVs from any of their studies?

Using our current approach, it is not possible to determine the levels of MV-lEVs to free sEV. Without analyzing serial ultrathin sections, determination of the relative ratio of MV-lEVs and sEVs would depend on the actual section plane. In future projects, we will determine the ratio of LC3 positive and negative sEVs by single EV analysis techniques (such as SP-IRIS). In the revised manuscript, additional TEM images are included to provide evidence for the simultaneous presence of sEVs and MV-lEVs and MV-lEVs both inside and outside of the circulation.

c. Please discuss the single versus double membrane issue (relating to experiments proposed above).

We discuss this question in more details in the revised manuscript.

d. Please point out that the release mechanism (plasma membrane budding) will involve different molecular mechanisms to establish exosome release, and this might provide a route to determine relative importance.

We are currently running a systemic analysis of the release mechanism of amphiectosomes, and this will be the topic of a separate manuscript.

**Reviewer #3 (Recommendations For The Authors):**
* The model is not supported.* The data is not of quality.* The appropriate methods are not exploited.

We are sorry, we cannot respond to these unsupported critiques.